# Physiological flexibility of phytoplankton impacts modeled chlorophyll and primary production across the North Pacific Ocean

Yoshikazu Sasai[1], Sherwood Lan Smith[1], Eko Siswanto[2], Hideharu Sasaki[3], and Masami Nonaka[3]

[1]Earth Surface System Research Center (ESS), Research Institute for Global Change (RIGC), Japan Agency for Marine-Earth Science and Technology (JAMSTEC), Yokosuka, Japan
[2]ESS,RIGC,JAMSTEC,Yokohama,Japan
[3]Application Laboratory (APL), Research Institute for Value - Added - Information Generation (VAiG), JAMSTEC, Yokohama, Japan

**Correspondence:** Yoshikazu Sasai (ysasai@jamstec.go.jp)

**Abstract.** Phytoplankton growth, and hence biomass responds to variations in light and nutrient availability in the near-surface ocean. A wide variety of models have been developed to capture variable chlorophyll:carbon ratios due to photoacclimation, i.e. the dynamic physiological response of phytoplankton to varying light and nutrient availability. Although photoacclimation models have been developed and tested mostly against laboratory results, their application and testing against the observed flexible response of phytoplankton communities remains limited. Hence the biogeochemical implications of photoacclimation in combination with ocean circulation have yet to be fully explored. We compare modeled chlorophyll and primary production from an inflexible phytoplankton functional type model (InFlexPFT), which assumes fixed carbon (C):nitrogen (N):chlorophyll (Chl) ratios, to a recently developed flexible phytoplankton functional type model (FlexPFT), which incorporates photoacclimation and variable C:N:Chl ratios. We couple each plankton model with a 3-D eddy-resolving ocean circulation model of the North Pacific and evaluate their respective performance versus observations (e.g., satellite imagery and vertical profiles of in-situ observations) of Chl, and primary production. These two models yield different horizontal and vertical distributions of Chl and primary production. The FlexPFT reproduces observed subsurface Chl maxima in the subtropical gyre, although it overestimates Chl concentrations. In the subtropical gyre, where light is sufficient, even at low nutrient concentrations, the FlexPFT yields higher chlorophyll concentrations and faster growth rates, which result in higher primary production in the subsurface, compared to the InFlexPFT. Compared to the FlexPFT, the InFlexPFT yields slower growth rates, and lower Chl and primary production. In the subpolar gyre, the FlexPFT also predicts faster growth rates near the surface, where light and nutrient conditions are most favorable. Compared to the InFlexPFT, the key differences that allow the FlexPFT to better reproduce the observed patterns are its assumption of variable, rather than fixed, C:N:Chl ratios and inter-dependent, rather than strictly multiplicative, effects of light- (photoacclimation) and nutrient- (uptake) limitation. Our results suggest that incorporating these processes has the potential to improve chlorophyll and primary production patterns in the near-surface ocean in future biogeochemical models.

# 1 Introduction

Marine phytoplankton carry out approximately half of global primary production (Field et al., 1998) and sustain the marine food web. Much effort has therefore been expended to understand and develop predictive models of phytoplankton growth and associated marine ecosystem processes and biogeochemistry. Phytoplankton models have for decades been constructed by combining various empirically-based formulations for different physiological processes, such as photosynthesis as a function of irradiance, growth as a function of nutrient availability, and the regulation of chlorophyll (Chl) content and cellular composition, which is termed photoacclimation (e.g., Platt and Jassby, 1976; Droop, 1983; Geider et al., 1998; Baklouti et al., 2006). Various formulations have been derived from laboratory experiments and in-situ observations, typically for the response of a single key process (e.g., nutrient uptake or growth rate) to one or a few key environmental variables (e.g., nutrient concentration, light intensity, temperature). Global ocean biogeochemical models require combining multiple processes with sufficient generality to apply over a wide range of environmental conditions. This formidable challenge can be approached in various ways, which are still debated (e.g., Flynn, 2003, 2010; Franks, 2009; Anderson, 2010; D'Alelio et al., 2016). For example, some models include numerous phytoplankton and zooplankton types (Ward et al., 2013), others resolve complexity selectively for specific trophic levels (Follows et al., 2007; Göthlich and Oschlies, 2012), and others incorporate physiological trade-offs into ecological parameterizations (Smith et al., 2016; Pahlow et al., 2020).

Most phytoplankton models as used in global ocean biogeochemical models (e.g., Follows et al., 2007; Totterdell, 2019), apply the Monod equation (Monod, 1949) for phytoplankton growth as a function of ambient nutrient concentration and assume fixed stoichiometry between carbon and nutrients in phytoplankton and organic matter (Redfield et al., 1963). However, the actual elemental composition of phytoplankton and organic matter varies depending on environmental conditions (e.g., Smith et al., 1992; Martiny et al., 2013; Garcia et al., 2018b; Liefer et al., 2019). In general, phytoplankton can sustain relatively high growth rates even when nutrient uptake rates are severely substrate-limited, by producing biomass containing less of whatever required elements are in short supply (Flynn, 2010). Phytoplankton grow with carbon, C: nitrogen, N ratios higher than the Redfield ratio when N is limiting and lower than the Redfield ratio when light is limiting (e.g., Goldman et al., 1979; Falkowski et al., 1985; Smith et al., 1992). The inability of the Monod equation to describe adequately laboratory experiments and in-situ observations for the dependence of growth rate on nutrient concentration led to the development of the Droop quota model (Droop, 1968, 1983). The Droop quota model for growth explicitly accounts for flexible stoichiometry by including independent state variables for carbon and nutrient biomasses with separate functions for acquiring each element (Caperon, 1968; Droop, 1968). Because the flexible stoichiometry of phytoplankton links phytoplankton growth to biogeochemical cycles, the model has been applied to 1-dimensional (1-D) and 3-dimensional (3-D) ocean biogeochemical models and proved useful to account for observations of the composition of phytoplankton and organic matter (e.g., Moore et al., 2001; Vichi et al., 2007; Ayata et al., 2013; Ward et al., 2013). However, the application of such detailed models at the global scale has been restricted by both practical considerations of their computational requirements and scientific concerns about increased complexity (e.g.,

a greater number of parameter values and processes) relative to simpler fixed stoichiometry models.

Recently, as a potential solution to this problem, Smith et al. (2016) derived a computationally efficient Instantaneous Acclimation (IA) approach, which represents flexible phytoplankton composition, similar to the Droop quota model, but without introducing additional state variables for each element or pigment considered. This "Flexible Phytoplankton Functional Type" (FlexPFT) model accounts for the acclimative response to changing light and nutrient conditions in terms of two trade-offs for allocation of intracellular response: (i) carbon versus nitrogen assimilation (Pahlow and Oschlies, 2013), and (ii) affinity for nutrient versus maximum uptake rate (Pahlow, 2005; Smith et al., 2009). Smith et al. (2016) applied the FlexPFT model in a 0-D setup, and assessed its performance in terms of phytoplankton seasonality, including variable composition in responses to changing light and nutrient conditions, at two observation sites (Station K2, 47°N, 160°E, and Station S1, 30°N, 145°E) in the North Pacific. Ward (2017) further tested this approach and suggested that it has promise for incorporating flexible stoichiometry into global ocean biogeochemical models. Kerimoglu et al. (2021) further assessed the performance of the IA approach, as compared to the typical assumption of fixed stoichiometry, in an idealized setup capturing typical seasonal variations of environmental conditions in a 1-D water column, accounting for the coupling of phytoplankton growth and biogeochemistry with physical transport by advection and diffusion. Anugerahanti et al. (2021) assessed the performance of the IA approach favorably compared to a suite of models of differing complexity, based on comparisons of 1-D model performance against extensive time-series observations from subtropical stations ALOHA (A Long term Oligotrophic Habitat Assessment, 22.45°N, 158°W) in the North Pacific and BATS (Bermuda Atlantic Time Series, 31.67°N, 64.167°W) in the North Atlantic. Masuda et al. (2021) applied the FlexPFT model in a global 3-D setup and showed that it could reproduce the global distribution of observed subsurface chlorophyll maxima (SCM). However, for 3-D applications, especially with global ocean biogeochemical models and earth system models, the challenge remains to reproduce the large-scale ocean conditions with computational efficiency and minimal tuning of model parameters (e.g., Masuda et al., 2021; Matsumoto et al., 2021). In this context, only limited tests have so far been conducted against oceanic observations. Here we explore the biogeochemical implications of the IA approach and the eco-physiological assumptions underlying the FlexPFT model in combination with large-scale ocean circulation.

Most biogeochemical models have a similar structure, with nitrogen as the main currency for a simplified food web, which generally includes phytoplankton and zooplankton, and a regeneration network with detritus, dissolved organic nitrogen, and various nutrients (i.e., Fasham et al., 1990). Whereas the more complex biogeochemical models have become more common (e.g., Follows et al., 2007; Totterdell, 2019), simple phytoplankton growth (fixed stoichiometry, without photoacclimation) models are still applied widely. In this study, we focus on the acclimative growth response of phytoplankton as incorporated in these models. To evaluate the performance and implications of this acclimative response of phytoplankton growth to varying light and nutrient conditions across the North Pacific Ocean, we compare modeled chlorophyll and primary production from an inflexible phytoplankton functional type model (InFlexPFT), which assumes fixed C:N:Chl ratios (fixed stoichiometry), to a recently developed phytoplankton model (FlexPFT, Smith et al., 2016), which incorporates photoacclimation and variable

C:N:Chl ratios. We apply these two phytoplankton models in a 3-D eddy-resolving ocean circulation model of the North Pacific, to assess each model's performance compared to observations of chlorophyll and primary production.

## 2 Methods and Materials

### 2.1 The Coupled Physical-Biological Model

We used a coupled physical-biological model of the North Pacific, consisting of a physical ocean model, which is an eddy-resolving ($1/10°$) OFES2 (the Ocean general circulation model For the Earth Simulator) that includes sea-ice (Masumoto et al., 2004; Komori et al., 2005; Sasaki et al., 2020) coupled with a simple nitrogen-based Nitrate-Phytoplankton-Zooplankton-Detritus (NPZD) pelagic model (Sasai et al., 2006, 2010, 2016). The OFES2 domain extends from $20°$S in the South Pacific to $68°$N in the North Pacific and from $100°$E to $70°$W. The OFES2 has $1/10°$ horizontal resolution with 105 vertical levels, from 5 m thickness at the surface to 300 m thickness at the maximum depth of 7500 m. The physical fields were spun up for 50 years under climatological forcing data (wind stresses, heat flux, and freshwater flux) from the Japanese 55-year Reanalysis (JRA55-do) (Tsujino et al., 2018) and from the initial condition of the observed climatological fields of temperature and salinity (World Ocean Atlas 2009, WOA09) (Antonov et al., 2010; Locarnini et al., 2010) without no motion for 50 years. After 50 years of spin-up integration, the OFES2 was forced by 3-hourly JRA55-do from 1958 to 1979. The last day of 1979 is used for the initial physical fields for performing the coupled physical-biological model simulation.

In the OFES2, an advection-diffusion equation, Eq. A1 (Appendix A), is used to calculate the evolution of four biological tracer concentrations (nitrogen-based units, mmol N m$^{-3}$): Nitrogen, $N$, Phytoplankton, $P$, Zooplankton, $Z$, and Detritus, $D$. The source and sink terms represent the biological activity (Eqs. A2 - A5) as described by Sasai et al. (2006, 2010, 2016). In this study, to examine how the physiological flexibility of phytoplankton impacts modeled chlorophyll and primary production (PP), two phytoplankton models (InFlexPFT and FlexPFT), respectively, are applied for the phytoplankton growth term, $\mu P$, in the Eq. A2. The remaining biological activity equations and biological parameters (e.g., Grazing, Mortalities of $P$ and $Z$, and Decomposition of $D$) are the same for both models (Appendix A), because we focus on the phytoplankton growth response and have ignored other processes (e.g., interactions between grazers and phytoplanktons, export and recycling). The initial nitrogen $N$ (mmol N m$^{-3}$) field is taken from the observed annual climatological values of WOA09 (Garcia et al., 2010). The initial $N$ concentration range from 5 to 20 (mmol N m$^{-3}$) in the subpolar surface and 0.1 to 5 (mmol N m$^{-3}$) in the subtropical surface. The initial phytoplankton and zooplankton concentrations are set to 0.2 mmol N m$^{-3}$ at the sea surface, decreasing exponentially with an e-folding scale depth of 100 m. Detritus is initialized to 0.1 mmol N m$^{-3}$ everywhere. These $P$, $Z$, and $D$ initial values are taken from Sasai et al. (2006, 2010, 2016). Two NPZD models are incorporated after the last day of 1979 of the physical fields in the OFES2. The two coupled physical-biological models are forced by 3-hourly JRA55-do from 1980 to 2019.

## 2.2 Formulations of Phytoplankton Growth in the Biological Model

Here, we briefly describe the two phytoplankton growth rate equations, InFlexPFT and FlexPFT, used in this study, which appear in the first term of the right-hand side, $\mu P$, in Eq. A2. In both models, phytoplankton growth rate, $\mu$ (day$^{-1}$), depends on irradiance, $I$, which is the intensity of photosynthetically active radiation calculated from daily mean shortwave radiation (W m$^{-2}$) of JRA55-do, nitrogen, $N$ (mmol N m$^{-3}$), and temperature, $T$ (°C). The phytoplankton growth rate equation is expressed by multiplying $N$ uptake, $I$-limitation, and $T$-limitation. $N$ and $T$ are used from the coupled physical-biological models' output. The InFlexPFT for growth rate, $\mu_{\text{IFL}}(N,I,T)$ (day$^{-1}$), is based on the Optimal Uptake kinetics equation (Smith et al., 2009):

$$\mu_{\text{IFL}}(N,I,T) = \mu_{max} \left( \frac{N}{N + \left( \frac{\hat{V}_0}{\hat{A}_0} \right) + 2\sqrt{\frac{\hat{V}_0 N}{\hat{A}_0}}} \right) S(I,T)F(T) \tag{1}$$

where $\mu_{max}$ is the potential maximum growth rate (day$^{-1}$), $\hat{V}_0$ is the potential maximum uptake rate for $N$ (day$^{-1}$), and $\hat{A}_0$ is the potential maximum affinity for $N$ (m$^3$ (mmol N)$^{-1}$ day$^{-1}$), following the Optimal Uptake equation (Smith et al., 2009, 2010) (Table 1). Compared to the Monod equation for growth as a function of ambient nutrient concentration, as typically applied in fixed composition models, this equation yields a similar response, but with a slightly flatter shape (Smith et al., 2009). $S(I,T)$ specifies the dependence on $I$ (Pahlow et al., 2013), and $F(T)$ is the Arrhenius-type temperature dependence. $S(I,T)$ and $F(T)$ are defined as, respectively:

$$S(I,T) = 1 - exp \left\{ \frac{-\alpha \hat{\theta} I}{\mu_{max} F(T)} \right\} \tag{2}$$

$$F(T) = exp \left\{ \frac{-E_a}{R} \left[ \frac{1}{T+298} - \frac{1}{T_{ref}+298} \right] \right\} \tag{3}$$

here $\alpha$ is the Chl-specific initial slope of growth versus light intensity (dimensionless, Table 1), and $\hat{\theta}$ is the Chl:C ratio (g chl (mol C)$^{-1}$) of the chloroplast, as described by Pahlow et al. (2013). In the InFlexPFT, the Chl:C ratio is set to a constant value ($\hat{\theta} = 0.6$), assuming fixed stoichiometry without photoacclimation. $E_a$ is the activation energy ($4.8 \times 10^4$ J mol$^{-1}$), which is set to a constant value, corresponding to a doubling of growth rate for a 10°C increase in temperature (i.e., $Q_{10} = 2.0$), which is a typical empirically-based value for the temperature sensitivity of phytoplankton growth rates (Eppley, 1972; Bissinger et al., 2008), $R$ is the gas constant (8.3145 J (mol K)$^{-1}$), and $T_{ref}$ is the reference temperature (taken as 20°C).

The FlexPFT assumes that optimally-based photoacclimation theory based resource allocation trade-off between light and nutrient (Pahlow et al., 2013; Smith et al., 2016). The growth rate, $\mu_{\text{FL}}(N,I,T)$ (day$^{-1}$) (Smith et al., 2016), is:

$$\mu_{\mathrm{FL}}(N,I,T) = \mu_{max} \left( 1 - \frac{Q_s}{Q(N,I,T)} - f_V(N,I,T) \right) S(I,T)F(T) \tag{4}$$

where $Q_s$ is the structural minimum cell quota (mol N (mol C)$^{-1}$) given as a fixed parameter (= $Q_0/2$, where $Q_0 (= 0.039)$ is the minimum cell quota, Edwards et al., 2012), $Q$ is the nitrogen cell quota, i.e. the intracellular $N$ content per unit carbon biomass (mol N (mol C)$^{-1}$) as a function of $I$, $N$, and $T$, and $f_V$ is the fractional allocation of intracellular resources to nutrient uptake (dimensionless) as defined by Pahlow et al. (2013). The cell quota, $Q(N,I,T)$, is:

$$Q(N,I,T) = Q_s \left[ 1 + \sqrt{ 1 + \left[ Q_s \left( \frac{\mu_{max} S(I,T)F(T)}{\hat{V}^N(N,T)} + \zeta^N \right) \right]^{-1} } \right] \tag{5}$$

where $\hat{V}^N(N,T)$ is the potential nutrient uptake rate (mol N (mol C)$^{-1}$ day$^{-1}$), and $\zeta^N$ is the energetic respiratory cost of assimilating inorganic nitrogen (0.6 mol C (mol N)$^{-1}$, Pahlow and Oschlies, 2013). The potential nutrient uptake rate, $\hat{V}^N(N,T)$, is:

$$\hat{V}^N(N,T) = \frac{\hat{V}_0 N}{N + \left( \frac{\hat{V}_0}{\hat{A}_0} \right) + 2\sqrt{\left( \frac{\hat{V}_0 N}{\hat{A}_0} \right)}} \tag{6}$$

The fractional allocation of intracellular resources to nutrient uptake, $f_V(N,I,T)$, as defined by Pahlow et al. (2013) is:

$$f_V(N,I,T) = \frac{\mu_{max} S(I,T)F(T)}{\hat{V}^N(N,T)} \left[ -1 + \sqrt{ 1 + \left[ Q_s \left( \frac{\mu_{max} S(I,T)F(T)}{\hat{V}^N(N,T)} + \zeta^N \right) \right]^{-1} } \right] \tag{7}$$

The differences between the two models are the trade-off between light and nutrient acquisition (Eqs. 5, and 7) and the variable Chl:C ratio ($\hat{\theta}$) in the light limitation term of Eq. 2, which are only included in the FlexPFT (Eq. 4) to account for the flexible response of phytoplankton growth to changing light and nutrient conditions (Eqs. 2, 5, 6, and 7). The optimal value of Chl:C ratio in the FlexPFT is applied when irradiance $I$ exceeds the threshold irradiance, below which the respiratory cost outweighs the benefits of producing chlorophyll (Pahlow et al., 2013; Smith et al., 2016). The InFlexPFT has the same nutrient uptake response as the FlexPFT (Eqs. 1 and 6), and the Chl:C ratio in the light limitation term (Eq. 2) is set to a constant value. The same temperature dependence, $F(T)$, is assumed in both models (Eq. 3). Parameter values, $\mu_{max}, \hat{V}_0, \hat{A}_0$, and $\alpha$ (Table 1) used in Eqs. 1 to 7 for the phytoplankton growth rate were tuned, separately for each coupled model, to confirm the reproducibility of the climatological seasonal variability of observed $N$, and Chl patterns in the near-surface of North Pacific. In this study, we have chosen to apply different parameter values, based on separate tuning of each coupled model (Table 1), in order to make a fair comparison of each model's ability to reproduce the climatological seasonal and spatial variability of N and Chl. Applying the same parameter values for both models would not be meaningful, given their different meanings within the different

growth equations. For example, the potential maximum growth rate, $\mu_{max}$, is 1.5 (day$^{-1}$) for the InFlexPFT, compared to 2.2 (day$^{-1}$) for the FlexPFT. Increasing the potential maximum growth rate decreases the surface $N$ concentration in the subpo-
lar gyre, to the point of depleting nutrient during summer, while increasing the surface Chl concentration across the whole gyre.

Compared to the InFlexPFT, the FlexPFT model yields different growth rates because it instantaneously optimizes both the allocation factor, $f_V(N, I, T)$ (Eq. 7) , and the Chl:C ratio of the chloroplast, $\hat{\theta}$, which appears in the light limitation term, $S(I, T)$ (Eq. 2). Therefore, it is possible to understand the modeled patterns of Chl (mg m$^{-3}$) and PP (mgC m$^{-3}$ day$^{-1}$)
over the North Pacific Ocean by comparing the expressions for growth rate, $\mu$ as a function of $I$, $N$, and $T$ for each model, respectively. Results are presented for simulated years 2000 to 2019 with verifiable observation data. For the InFlexPFT, Chl concentration (mg m$^{-3}$) is $P$ (mmol N m$^{-3}$) $\times$ the constant Chl:N ratio (1.59 g Chl (mol N)$^{-1}$), and PP (mgC m$^{-3}$ day$^{-1}$) is $\mu_{IFL}P$ (mmol N m$^{-3}$ day$^{-1}$, Eq. 1) $\times$ the fixed C:N ratio (Redfield ratio = 106:16 mol C (mol N)$^{-1}$). In the FlexPFT, the Chl concentration (mg m$^{-3}$, $= P \times \hat{\theta}/Q$) is the phytoplankton concentration, $P$ (mmol N m$^{-3}$), $\times$ the variable Chl:N ratio (g
Chl (mol N)$^{-1}$, $\hat{\theta}/Q$), and Primary Production, PP (mgC m$^{-3}$ day$^{-1}$), is $\mu_{FL}P$ (mmol N m$^{-3}$ day$^{-1}$, Eq. 4) $\times$ the variable C:N ratio (mol C (mol N)$^{-1}$), $1/Q$) (Eq. 5 and Smith et al., 2016).

## 2.3   Observational Data

The last 20 years (2000-2019) average of model results were compared with satellite data, in-situ observations, and the cli-matological data (Chl, nitrate, and temperature) to investigate the large scale variation over the North Pacific. Although the
model and observation periods differ somewhat, using the satellite and in-situ observation data observed during the sim-ulation period (2000s), we compare whether the horizontal and vertical patterns of climatological seasonal variations can reproduce the patterns captured by the satellite and the snapshot observations. Especially, we focused on the Chl and PP patterns, which strongly reflect effects of the different assumptions about how growth rates depend on light and nutrients. Sea surface Chl satellite imagery is derived from the Moderate Resolution Imaging Spectroradiometer (MODIS) - Aqua
(http://doi.org/10.5067/AQUA/MODIS/L3M/CHL/2022), using the seasonal climatological (averaged from 2003 to 2019) data of Level-3 global browser. Ship observed Chl data along the two sections (north-south and east-west) in the North Pacific are available from the websites of the Japan Meteorological Agency, JMA (https://www.data.jma.go.jp), and Japan Oceanographic Data Center, JODC (https://www.jodc.go.jp), respectively. Along the north-south section (165°E) in the western North Pa-cific, JMA research vessels have observed regularly from 2005 to the present. Along the east-west section (around 35°N)
in the central North Pacific, observations were conducted in summer 2002 to 2003, and published by the JODC. Seasonal climatological nitrate and temperature distributions are acquired from the World Ocean Atlas 2018 (WOA18) (Garcia et al., 2018a; Locarnini et al., 2018). The PP data sets are available at three time-series stations: Stations K2 (47°N,160°E) and S1 (30°N,145°E) in the western North Pacific, as implemented by the K2S1 project (Matsumoto et al., 2014, 2016; Honda et al., 2017) (https://ebcrpa.jamstec.go.jp/k2s1/en/index.html), and Station ALOHA (22.45°N, 158°W) in the central North Pacific as
operated under the Hawaii Ocean Time series (HOT) program (Karl et al., 1996, 2021) (https://hahana.soest.hawaii.edu/hot/).

## 3 Results

This section assesses the models' performance and examines the impact of physiological flexibility on modeled Chl and PP by comparing the results of two coupled physical-biological (InFlexPFT and FlexPFT) models against MODIS-Aqua imagery and vertical profiles of in-situ observations (JMA and JODC ship observation lines). Modeled phytoplankton is controlled by various ecological processes (e.g., grazing, mortality, export and recycling), but in this study, we focus on the implications of specific assumptions about how phytoplankton growth rate depends on light, nutrient, and temperature. The eddy-resolving ocean circulation model (OFES2) has fine horizontal resolution ($1/10°$, about 10 km), and reproduces the western boundary current, Kuroshio, the observed variability in the Kuroshio Extension region between the subtropical and subpolar gyres, mesoscale eddies, and upwelling events (e.g., Masumoto et al., 2004; Sasai et al., 2010; Sasaki et al., 2020). In addition, the seasonal variability of $T$ and $N$ fields in the near-surface over the North Pacific are also well reproduced (not shown). These physical processes directly or indirectly affect the nutrient and light environments, and biogeochemical processes (e.g., Oschiles, 2002; Gruber et al., 2011; Levy et al., 2014; Sasai et al., 2010, 2019), and are important for supplying the nutrients needed by phytoplankton, especially in the coastal upwelling regions and the oligotrophic subtropical gyre. Here we focus on the different assumptions about how phytoplankton growth rate depends on ambient nitrogen concentration and light intensity. First, the reproducibility of seasonal and horizontal Chl distributions is described. As the Chl concentration in the FlexPFT is calculated from $P \times \hat{\theta}/Q$, and reflects the changes in $\hat{\theta}$ and $Q$, we examine how variations in C:N:Chl ratios impact the surface Chl pattern. Next, we compare the results of the two coupled physical-biological models in terms of Chl and PP along two vertical transects (north-south and east-west, respectively) in the North Pacific, and discuss the reasons for the differences. Especially, the role of photoacclimation in the formation of SCM and the growth rate on the variable C:N:Chl ratios of phytoplankton. Finally, the difference in PP as calculated by these two models over the North Pacific and the comparison with limited PP vertical profiles are discussed. The extent to which the different growth rate (InFlexPFT vs FlexPFT) affects the estimated PP is described.

### 3.1 Comparison of Surface Chl Patterns

Fig. 1 shows the surface Chl distribution as simulated over the North Pacific by the two models, compared with MODIS-Aqua imagery to assess the two models' performance. Overall, using an eddy-resolving ($1/10°$) OFES2, the two models reproduce the climatological seasonal variations of surface Chl pattern between the subtropical ($< 0.2$ mg m$^{-3}$, blue shaded) and subpolar ($> 0.2$ mg m$^{-3}$, green and yellow shaded) gyres, as captured by the MODIS-Aqua imagery. In particular, the contrast between two gyres and the coastal upwelling region more clearly than lower-resolution (e.g., $1°$, about 100 km) models (e.g., Moore et al., 2001; Vichi et al., 2007; Follows et al., 2007; Göthlich and Oschlies, 2012) by using the OFES2. In addition, both models reproduce the boundary ($0.2$ mg m$^{-3}$) between the subtropical and subpolar gyres, its seasonal variations, and high concentration ($> 0.4$ mg m$^{-3}$) in the coastal upwelling regions (California coast) as seen in the MODIS-Aqua imagery. Compared to the InFlexPFT, the FlexPFT model produces greater variations and steeper horizontal gradients of Chl concentration. Especially in the open ocean north of $30°$N, coastal upwelling regions off the western coast of North America, the Sea of Okhotsk, and

the Bering Sea, the FlexPFT produces higher surface Chl ($> 0.4$ mg m$^{-3}$) than the InFlexPFT, but underestimates Chl in the subtropical gyre (south of $30°$N, $< 0.1$ mg m$^{-3}$). The FlexPFT also reproduces greater seasonal variation of surface Chl in the subpolar gyre, similar to the pattern in the MODIS-Aqua imagery. On the other hand, in the subtropical gyre, the InFlexPFT is more similar to the MODIS-Aqua imagery. These differences in the spatial and seasonal distributions of surface Chl result from the different phytoplankton growth rate equations (Eqs. 1 and 4). The Chl:N ratio is calculated from $\hat{\theta}$ (photoacclimation) and $Q$ (nitrogen cell quota, Eq. 5), and especially the difference between the FlexPFT's variable Chl:N ratios in 0.1 to 3.0, versus the fixed value of 1.59 in the InFlexPFT. In the subtropical gyre, the FlexPFT's Chl:N ratio is low ($< 0.6$), and the Chl concentration is underestimated compared with the MODIS-Aqua imagery. In the subpolar gyre and coastal upwelling regions, the FlexPFT's Chl:N ratio is high ($> 1.0$), and the Chl concentration is close to the MODIS-Aqua imagery.

## 3.2 Comparison of Vertical Distributions of Chl and PP along the Two Transect Lines

The limited data available from observations made by research vessels do capture key spatial and temporal distributions. Here, the reproducibility of the vertical distribution of Chl for each model is discussed along two observation lines over the North Pacific Ocean: a north-south section along $165°$E in Fig. 2, and an east-west section around $35°$N in Fig. 3. In particular, we focus on the SCM formed in summer, and discuss the effects of different assumptions about how phytoplankton growth rate (Eqs. 1 and 4) depends on $N$ concentration and light intensity, $I$, as well as their effects in combination with temperature, $T$ (Figs. 4, 5, 6, and 7).

JMA research vessels have made observations on the north-south line along $165°$E regularly since 1997, with good seasonal coverage. The best data coverage for vertical Chl profiles is available for the summer of 2006. The observed SCM ($> 0.1$ mg m$^{-3}$) depth varies from 50 m near the equator to 150 m in subtropical regions, and the FlexPFT clearly reproduces the observed pattern of SCM near the nutricline (close to 1 mmol N m$^{-3}$) along the $165°$E line (Fig. 2), with simulated values close to the observed SCM (Figs. 2a and 2c). However, the FlexPFT underestimates near-surface Chl ($< 0.02$ mg m$^{-3}$, dark blue shaded). By contrast, the InFlexPFT (Fig. 2b) cannot reproduce the observed SCM even though its modeled distributions of $N$ and $T$ are similar to the corresponding observations (Figs. 2d, 2e, 2f, 2g, and 2h). The $N$ and $T$ distributions reproduced in the OFES2 are mainly controlled by the physical processes, and the difference in vertical Chl distributions is influenced by the difference in the response of the phytoplankton growth to the light and nutrient conditions (Eqs. 1 and 4). The FlexPFT, which incorporates the photoacclimation and the trade-off between light and nutrient acquisition, reproduces the observed vertical Chl distributions much better than the InFlexPFT, especially, the depth and structure of SCM.

The vertical distribution of Chl also varies with longitude along the east-west transect around the boundary ($35°$N) between the subpolar and subtropical gyres (Fig. 3). The limited JODC Chl observations (Summer, 2001-2002) span the east and west of the North Pacific. The observed SCM ($> 0.1$ mg m$^{-3}$) appears between 50 and 150 m depth and deepens to the east (Fig. 3a), following the distribution of nutricline depths (close to 1 mmol N m$^{-3}$) (Fig. 3d). Compared to the north-south transect,

here the near-surface $T$ gradient is not as steep (Figs. 2 and 3), with $T$ near the surface decreasing from the center to the east and west (Fig. 3g). The model reproduces the observed $T$ distribution, and modeled $N$ distribution is similar to the observed data (Figs. 3e, 3f, and 3h). The FlexPFT clearly reproduces the observed SCM around the nutricline depth (Figs. 3c and 3f) from east to west, whereas the InFlex does not (Fig. 3b). On the eastern side of the North Pacific, both models have the deep nutricline depth, and the FlexPFT underestimates the observed Chl (Figs. 3a and 3c) and $N$ (Figs. 3d and 3f).

To clarify the mechanistic reasons for differences in the vertical distributions of Chl between the two models (Figs. 2 and 3), the vertical distributions of models' PP (mgC m$^{-3}$ day$^{-1}$), and related phytoplankton growth rate (day$^{-1}$) from Eqs. 1 and 4, and the variable C:N ratio (reciprocal of the $N$ quota of phytoplankton carbon ratio, $1/Q$, Eq. 5) in the FlexPFT, are shown in Figs. 4 and 5. The PP was calculated from the phytoplankton growth rate and the C:N ratio, which for the FlexPFT is variable (Figs. 4e and 5e) whereas it is constant for the InFlexPFT. In summer (Figs. 2 and 3), the SCM is clearly formed in the subsurface layer except for the subpolar gyre (north of 40°N). The vertical distributions of PP and phytoplankton growth rate form maxima along the nutricline depth. Both models predict high PP ($> 10$ mgC m$^{-3}$ day$^{-1}$) near the surface in the subpolar regions along 165°E, with minimal PP ($< 1$ mgC m$^{-3}$ day$^{-1}$) near the bottom of the euphotic layer, which is close to 100 m depth (Figs. 4a and 4b). Overall, compared to the InFlexPFT, the FlexPFT produces greater PP, with profiles extending deeper into the subsurface (100 m depth) to the south of 30°N. Both models produce fast growth rates in the subpolar surface layers ($> 0.5$ day$^{-1}$) and the subtropical subsurface layers ($> 0.2$ day$^{-1}$) (Figs. 4c and 4d). Especially in the subtropical subsurface layers, the FlexPFT predicts much faster growth than the InFlexPFT, and this difference in phytoplankton growth rate is reflected in the vertical distributions of Chl (Figs. 2b and 2c) and PP (Figs. 4a and 4b). The variable C:N ratio in the FlexPFT (Fig. 4e) also contributes substantially to its greater PP values (Fig. 4d), compared to the constant C:N ratio in the InFlexPFT (Fig. 4c). In Fig. 4e, the C:N ratio in the FlexPFT is high ($> 20$) near the surface around the equator and subtropical regions, and near the Redfield ratio (106:16 = 6.625) elsewhere in the subpolar region and below the euphotic layer (below 100 m depth). Our results show that, compared to the assumption of constant C:N:Chl ratios, as in the InFlexPFT, accounting explicitly for variable C:N (4.0 to 25.0 in Fig. 4e) ratios and the acclimated growth response of phytoplankton, as in the FlexPFT, yields substantially better reproduction of Chl and PP profiles within the euphotic layer.

As in Fig. 4, the vertical distributions of modeled PP (mgC m$^{-3}$), related phytoplankton growth rate (day$^{-1}$) from Eqs. 1 and 4, and the variable C:N ratio in the FlexPFT (Eq. 5) in the east-west section are shown in Fig. 5. Both models predict high PP ($> 8$ mgC m$^{-3}$ day$^{-1}$) between 50 m and 100 m, with shallower distributions on the west side and deeper toward the east (Figs. 5a and 5b). Compared to the InFlexPFT, the FlexPFT predicts higher PP ($> 10$ mgC m$^{-3}$ day$^{-1}$), especially in the subsurface (near 50 m depth). Although both models predict faster growth on the west side ($> 0.5$ day$^{-1}$), decreasing towards the east ($< 0.2$ day$^{-1}$), their patterns differ (Figs. 5c and 5d). In the InFlexPFT, the phytoplankton growth rate is fastest near the surface and decreases with depth. On the other hand, the FlexPFT predicts increasing growth rate with depth from the surface to intermediate depths. Both models produce subsurface maxima in PP (Figs. 5a and 5b), with a stronger pattern for the FlexPFT model. The FlexPFT predicts an even stronger subsurface maximum in the vertical distribution of Chl (Fig. 3c).

This results from the FlexPFT's combination of photoacclimation and variable C:N ratio, which is consistently high ($> 10$) in the euphotic layer (0 - 100 m) (Fig. 5e). Below the euphotic layer and to the west side, the FlexPFT's C:N ratio approaches the Redfield ratio (106:16 = 6.625), and therefore the modeled Chl and PP differ less between the two models.

To examine the inter-dependent impacts of $N$ concentration and light intensity, $I$ on the phytoplankton growth rate in the FlexPFT, as compared with their simple multiplicative dependencies in the InFlexPFT, Figs. 6 and 7 show scatter diagrams of modeled phytoplankton growth rate and variable C:N ratio ($1/Q$) versus $N$ concentration (mmol N m$^{-3}$) at the three locations along 165°E in Fig. 4 and around 35°N in Fig. 5. The difference in depth (0 m, 50 m, and 100 m) corresponds to the light intensity at each location (strong on the surface and weaker at 100 m depth). Compared to the InFlexPFT, the FlexPFT maintains faster growth as either light or nutrient becomes limiting everywhere. The phytoplankton growth rate equations (Eqs. 1 and 4) also depends on temperature, $T$ (Eq. 3), as shown by colors. The explicitly formulated temperature dependence of the growth rate is the same in both models. In general, $I$ and $T$ both depend on latitude, and both decrease with increasing depth. At 25°N latitude (Figs. 6a and 6d), in the subtropical gyre, both models predict phytoplankton growth rates exceeding 0.2 (day$^{-1}$) near the surface, despite the low $N$ concentrations ($< 0.1$ mmol N m$^{-3}$), high $I$ and $T$ enhance the growth rate. Compared to the InFlexPFT, the FlexPFT maintains faster growth rates at the surface and 50 m because it dynamically allocates intracellular resources to cope with variations in $N$ concentration and $I$ (Smith et al., 2016, and Eq. 4). At the boundary between the subtropical and subpolar gyres, which lies approximately between 35°N and 40°N latitude, the two models predict clearly different patterns of phytoplankton growth rate compared to those at 25°N (Figs. 6b, 6c, 6e, and 6f). At the gyre boundary and in the subarctic, both models tend to produce faster growth rates at 50 m, where the availability of light and nutrients is better balanced to support phytoplankton growth, compared to the surface and 100 m depth. Compared to temperature, nutrient- and light-limitation exert greater control over the modeled growth rates with both models, but more so for the FlexPFT. Nutrient limitation is the strongest determinant of growth rates at the surface and intermediate (50 m) depth, whereas light limitation strongly suppresses growth rates (and their range of variability) at 100 m depth.

Similar to Fig. 6, Fig. 7 shows modeled phytoplankton growth rates at three selected locations along the 35°N latitude transect (Fig. 5). The two locations (160°E, 36°N, and 170°E, 34°N) on the west side of the North Pacific are close to (165°E, 35°N) in Fig. 6, and have similar characteristics. At these locations, near the boundary of the subtropical and subpolar gyres, the nutricline depth is shallower than on the eastern side. The InFlexPFT at the surface and 50 m depth shows the two curves of phytoplankton growth rate depending on the $N$ concentration, $I$, and $T$ (Figs. 7a and 7b) similar to Fig. 6b. Again, the InFlexPFT predicts the fastest growth rates at the high $T$ ($> 20$°C), despite low $N$ concentrations. On the other hand, the FlexPFT's growth rates are more clearly related to ambient $N$ concentration with a less apparent relationship to $T$, despite assuming the same $T$ dependence in both models (Figs. 7d and 7e). As in Fig. 6, compared to the InFlexPFT, the FlexPFT predicts faster phytoplankton growth rates, with maximal growth at intermediate $N$ concentrations (despite lower $T$ compared to the surface), and wider variation of growth rate. At the eastern-most location (170°W, 30°N) shown in the right-most column of Figs. 7c and 7f, the patterns are more similar to the subtropical area shown in the left-most column of Figs. 6a and 6d, albeit

with somewhat greater variability of growth rate for both models near the surface, where temperatures are high and nutrients scarce. As at the subtropical location (25°N latitude) shown in the left-most column of Fig. 6, at this location the InFlexPFT also exhibits a stronger apparent temperature sensitivity (hence a wider range of growth rates) than the FlexPFT, but here only near the surface. Neither model shows a strong apparent relationship between growth rate and $N$ concentration at this location, indicating that light, nutrients, and temperature all substantially limit the growth here.

Compared to the InFlexPFT, the FlexPFT produces higher PP because its variable C:N ratio (4.0 to 25.0) substantially exceeds the Redfield ratio (106:16 = 6.625), with a consistent increase in C:N ratio from intermediate depths to the surface everywhere (Figs. 6 and 7). Where the phytoplankton growth rate reaches maximal values at 35°N and 40°N, the C:N ratios are between 10 and 15 (Figs. 6h, 6i, 7g, and 7h), which enhances PP (Figs. 4b and 5b). Along the east-west transect as well (Fig. 7), the FlexPFT again predicts maximal C:N ratios near the surface ($> 20$), where nutrient concentrations are lowest and light (and temperature) levels are highest, with a similar range of C:N ratios as along the north-south transect (Fig. 6). On the west side, the C:N ratio in the FlexPFT (Figs. 7g and 7h) exceeds the Redfield ratio ($> 6.625$), except below the euphotic layer, and growth rates are maximal at intermediate nutrient concentrations and light intensities. On the other hand, on the east side, where ambient nutrient concentrations are consistently low, despite the high C:N ratio in the FlexPFT ($> 20$ in Fig. 7i), modeled growth rates differ little between the two models.

Overall, at the surface and 50 m depth, the modeled patterns of growth rate versus $N$ concentration are more clearly separated with the FlexPFT model, which produces a steeper and more consistent increase in growth rate from low to intermediate $N$ concentrations, compared to the InFlexPFT. These variations in the locally realized maximum growth rate result from the FlexPFT's growth optimization scheme, which re-allocates intracellular resources, resulting in an inter-dependent response to $I$ and $N$ availability (Fig. 3 of Smith et al., 2016, and Eq. 4). By contrast, the InFlexPFT's simple multiplicative dependencies on $I$ and $N$ concentration result in a steeper decrease in growth rate as either resource becomes limiting. Despite the assumption of the same inherent $T$ sensitivity for both models, the optimal resource allocation thus results in a weaker apparent relationship between modeled growth rates and ambient $T$ in the FlexPFT model, which predicts the fastest growth rates at intermediate $N$ concentrations despite lower $T$ than at the surface. By contrast, the InFlexPFT model, because of its weaker dependence on nutrient concentration and light, predicts the fastest growth rates at the highest temperatures, with a stronger apparent relationship between growth rate and ambient temperature.

### 3.3 Vertical Profiles of PP at Three Stations and PP Patterns

Fig. 8 shows a direct comparison of modeled daily mean and observed vertical profiles of PP, for three time-series stations in the North Pacific: Station K2 in the western subpolar gyre, and Stations S1 and ALOHA in the western and eastern subtropical gyre, respectively. At Station K2, observed PP is more variable above 25 m depth, and less variable below 25 m depth (Fig. 8a). Overall, compared to the FlexPFT (8 to 50 mg C m$^{-3}$ day$^{-1}$), the InFlexPFT predicts less temporal variability of PP (5 to

15 mg C m$^{-3}$ day$^{-1}$), except below 50 m depth at Station K2, where the FlexPFT predicts faster growth than the InFlexPFT (Figs. 4c and 4d). At Station S1, the temporal variation of observed PP is greater above 50 m depth, with less variability below 50 m depth (Fig. 8b). Compared to Station K2 (subpolar gyre), the highest PP occurs deeper at Station S1 (subtropical gyre), where light limitation is more substantial, which suggests that PP could be enhanced by an increase in light levels despite the relatively low $N$ concentrations. Compared to the InFlexPFT (1 to 8 mg C m$^{-3}$ day$^{-1}$), the FlexPFT shows greater temporal variation of PP above 75 m depth (3 to 50 mg C m$^{-3}$ day$^{-1}$). This difference depends on whether the phytoplankton growth ratio calculated by the model can reflect the optimal $N$ and light environment (FlexPFT) or not (InFlexPFT). Unlike the two stations in the western North Pacific, at Station ALOHA, both models underestimate the observed PP (Fig. 8c). The observed PP from the surface to 100 m depth is large, and PP near the surface is about the same as at the western Station S1. The FlexPFT underestimates the mean PP (2 mg C m$^{-3}$ day$^{-1}$), but the temporal variation (1 to 10 mg C m$^{-3}$ day$^{-1}$) is closer to the observed variability than with the InFlexPFT ($< 1$ mg C m$^{-3}$ day$^{-1}$). Overall, compared to the InFlexPFT, the FlexPFT agrees better with the observed vertical PP profiles and its temporal variability.

The vertical distribution of PP differs along the two transect lines (north-south and east-west, respectively) (Figs. 4 and 5). To investigate the models' PP variations within the euphotic layer, Fig. 9 shows the seasonal variations of vertically integrated PP from the surface to 100 m depth for the two models. The InFlexPFT produces weak seasonality for PP (Figs. 9a, 9b, 9c, and 9d), with maximal values in spring ($> 800$ mg C m$^{-2}$ day$^{-1}$) at the boundary between the subtropical and subpolar gyres (Fig.9b), because the spring bloom occurs both horizontally and vertically (e.g., Fig. 1f). At the gyre boundary, in addition to the surface, primary production is greater compared to other regions. In this region, the nutricline depth (close to the base of the euphotic layer) and the light intensity are optimal for the spring production. Along the east-west transect, from 30°N to 40°N, PP varies with $N$ concentration, which is high on the west side and low on the east side (Figs. 3d, 3e, and 3f). In summer and fall, when $N$ concentration in the subtropical gyre decreases in the euphotic layer, even where light intensity, $I$, is sufficient, $N$ availability limits the phytoplankton growth rate (Eq. 1), so that PP ($< 400$ mg C m$^{-2}$ day$^{-1}$) does not increase (Figs. 9c and 9d). On the other hand, in the subpolar gyre, the PP ($> 400$ mg C m$^{-2}$ day$^{-1}$) in winter, summer, and fall increase with $N$ (Figs. 9a, 9c, and 9d). In the coastal upwelling regions, $N$ supplied from below the euphotic layer sustains higher PP ($> 800$ mg C m$^{-2}$ day$^{-1}$) compared to the open ocean. The FlexPFT predicts wider seasonal variations of PP across the North Pacific (Figs. 9e, 9f, 9g, and 9h), which are especially noticeable at the boundary between the subtropical and subpolar gyres, along the Kuroshio Current flowing south of Japan, and in the coastal upwelling region off California. Compared with the seasonal variations of PP in the InFlexPFT, the FlexPFT's growth rate and the variable C:N ratio have a great influence on the spatiotemporal variations of PP (Figs. 4, 5, and 8). The FlexPFT's springtime PP reaches twice ($> 1600$ mg C m$^{-2}$ day$^{-1}$) that of the InFlexPFT (Figs. 9b and 9f). In the subtropical gyre, where the light intensity in the euphotic layer is sufficient for phytoplankton growth (Eq. 4), $N$ concentration in winter is higher than in summer and fall, and as a result, PP values are relatively high ($> 800$ mg C m$^{-2}$ day$^{-1}$), although still lower than during spring (Fig. 9e). In the subpolar gyre, where $N$ concentrations are high, as the light environment improves from winter to summer, PP increases ($> 1000$ mg C m$^{-2}$ day$^{-1}$) within the euphotic layer (Figs. 9g and 9h). In the coastal upwelling regions and the Kuroshio Extension, the FlexPFT, because

of its optimal resource allocation, predicts much greater PP ($> 1600$ mg C m$^{-2}$ day$^{-1}$) than the InFlexPFT.

The difference in phytoplankton growth rate between the two models is also reflected in the spatiotemporal distribution of PP (Figs. 8, and 9), similar to Chl distribution (Figs. 1, 2, and 3). These differences are greatest in the boundary region between the subpolar and subtropical gyres, the subpolar gyre and the coastal upwelling region. Compared to the stations of observed time series, the FlexPFT reproduces greater temporal variations than the InFlexPFT, which is more consistent with the observed variability. These results indicate that the FlexPFT (which incorporates photoacclimation and variable C:N:Chl ratios) better captures seasonal changes within the euphotic layer and near the nutricline than the InFlexPFT. Estimated primary production in the North Pacific basin (20°N - 60°N, 130°E - 110°W) from Fig. 9 with the FlexPFT is 5.0 - 5.6 PgC yr$^{-1}$, and with the InFlexPFT is 2.3 - 2.5 PgC yr$^{-1}$ over the simulated period of 2000 - 2019, respectively, and the FlexPFT's estimate is about twice that of the InFlexPFT. Although not directly comparable to our estimates, the global primary production as estimated by the satellite and global biogeochemical models remains large: from 38.8 - 42.1 PgC yr$^{-1}$ over the period of 1998 - 2018 (Kulk et al., 2020) and from 38 - 79 PgC yr$^{-1}$ (Carr et al., 2006).

## 4 Discussion

These spatial and seasonal differences in modeled chlorophyll and primary production patterns result from the different underlying assumptions about how phytoplankton respond to changing conditions. As found in previous studies (Anugerahanti et al., 2021; Kerimoglu et al., 2021; Masuda et al., 2021), our results show that photoacclimation and variable C:N:Chl ratios, as represented by the FlexPFT, are important for capturing observed distributions of PP and especially the SCM. For reproducing the latter, capturing the unimodal distribution of Chl:C ratio over depth is particularly important (Chen and Smith, 2018; Kerimoglu et al., 2021).

As a characteristic vertical profile, the SCM varies seasonally and across ocean regions (Cullen, 1982), driven by different light attenuation levels and nutricline depths. Various mechanisms are known and purported to contribute to SCM formation, as reviewed by Cullen (2015). These mechanisms include variations in phytoplankton growth rate along the vertical gradients of light and nutrient availability, physiologically controlled behaviors such as swimming or buoyancy regulation, and photoacclimation of pigment content. It is impossible to capture the vertical profiles of Chl with satellite observations, and it is therefore important to verify the SCM field reproduced by the model using in-situ observations (e.g., Shulenberger and Reid, 1981; Furuya, 1990). Using a 3-D biogeochemical ocean model coupled with the same FlexPFT model, Masuda et al. (2021) showed that the observed global scale SCM distribution can be reproduced by incorporating photoacclimation in response to varying nutrient and light conditions. Various mechanisms likely contribute differently in different oceanic regimes and other mechanisms are important for reproducing specific features of SCM (e.g., Moeller et al., 2019; Wirtz and Smith, 2020). Moeller et al. (2019) proposed a new mechanism, which is light-dependent grazing by microzooplankton reduces phytoplankton biomass near the surface but allows accumulation at depth, for SCM formation. Furthermore, vertical migration by phytoplankton can

explain the occurrence of SCM consistently above the nutricline depth, which photoacclimation alone cannot (Wirtz and Smith, 2020). Wirtz et al. (2022) also suggested that phytoplankton vertical migration fuels up to 40% (>28 tg yr$^{-1}$ N) of new production and directly contributes 25% of total oceanic net primary production, which thier modeling study estimated at 56 PgC yr$^{-1}$.

Observed elemental ratios of phytoplankton and particulate organic matter in surface layers deviate substantially from the Redfield ratio (e.g., Goldman et al., 1979; Garcia et al., 2018b; Liefer et al., 2019). Molar C:N ratios of phytoplankton vary from 4 to 60 (mol C: mol N) between phylogenetic groups. However, our current implementation FlexPFT model includes only one phytoplankton type, which varies its C:N ratio as it acclimates to the varying light, nutrient, and temperature levels. Sauterey and Ward (2022) found that nitrogen and temperature mostly determined variations in the C:N ratio of phytoplankton, with variable contributions from other factors across the North Atlantic. By contrast, we find that with the FlexPFT model applied to the North Pacific, nitrogen and light levels are the primary determinants of C:N:Chl ratios, with temperature playing a lesser role. Accurate modelling of Chl and PP patterns requires accounting for these dependencies to resolve the variable elemental composition and pigment content of phytoplankton. This may be important with respect to food quality effects (e.g., Kwiatkowski et al., 2018; Matsumoto et al., 2020). That is, variations in the C:N ratio of phytoplankton are likely to impact trophic transfer and production by higher trophic levels. Although we do not address this here, it could make a good suggestion for future work.

## 5 Conclusions

We have focused on different formulations for phytoplankton growth response as represented in the biogeochemical models, and investigated the acclimation of phytoplankton growth to changing light, nutrient, and temperature conditions over the North Pacific using two physical-biological models. We compared modeled Chl and primary production from a recently developed phytoplankton model, FlexPFT (Smith et al., 2016), which incorporates photoacclimation and variable C:N:Chl ratios for changing light and nutrient conditions, and the InFlexPFT, which lacks photoacclimation and assumes constant C:N:Chl ratios (fixed stoichiometry). Our comparison of model results against summertime observations from the North Pacific revealed that, compared to the InFlexPFT, the incorporation of photoacclimation and variable C:N:Chl ratios for changing light, nutrient, and temperature conditions in the FlexPFT yields improved reproduction of observed Chl and primary production distributions in the near-surface ocean. The assumption of Instantaneous Acclimation (IA) allowed computationally efficient modeling of flexible phytoplankton composition and growth response in our basin-scale model, and IA is likely to be of even greater benefit for global-scale biogeochemical models. In the future, we plan to assess model performance for other seasons, years, and decadal variations, and to apply this approach in a global ocean biogeochemical model in order to examine the response of Chl and primary production to climate change. In addition, we will proceed with research on introducing flexible physiology to the growth of multiple phytoplankton, as well as associated food quality effects on predation by zooplankton, and the uncertainty

of other biological processes, such as nitrification, grazing, mortality, export, and recycling.

*Data availability.* The OFES2 coupled NPZD model simulations are available upon request. Observed data used in this study are available at the following sites: The surface Chl satellite imagery of MODIS-Aqua data is available for download at http://doi.org/10.5067/AQUA/MODIS/L3M/CHL/2. The vertical observed Chl data are available for download at Japan Meteorological Agency (https://www.data.jma.go.jp) and Japan Oceanographic Data Center (https://www.jodc.go.jp). WOA09 and WOA18 are available at https://www.ncei.noaa.gov (Antonov et al., 2010; Garcia et al., 2010; Locarnini et al., 2010; Garcia et al., 2018a; Locarnini et al., 2018). The primary production data sets at Stations K2 and S1, and Station ALOHA are available for download at https://ebcrpa.jamstec.go.jp/k2s1/en/index.html (Matsumoto et al., 2014, 2016; Honda et al., 2017) and https://hahana.soest.hawaii.edu/hot/ (Karl et al., 1996, 2021), respectively.

## Appendix A: NPZD model

The NPZD model in the OFES2 is defined as nitrogen, $N$, phytoplankton, $P$, zooplankton, $Z$, and detritus, $D$, based on Sasai et al. (2006, 2010, 2016). The evolution of biological tracer concentration, $B_i$, is governed by an advective-diffusive-reactive equation.

$$\frac{\partial B_i}{\partial t} = -\nabla \cdot (uB_i) + K_h \nabla^2 B_i + \frac{\partial}{\partial z}\left(K_z \frac{\partial B_i}{\partial z}\right) + sms(B_i) \tag{A1}$$

where $u$ is velocity vector of OFES2, $K_h$ is the lateral diffusion coefficient and $K_z$ is the vertical diffusion coefficient in the OFES2, and $sms(B_i)$ is the source-minus-sink (sms) term due to biological activity rate (mmol N m$^{-3}$ day$^{-1}$) in the NPZD model. For the individual tracers ($P$, $Z$, $D$ and $N$), the sms terms are given by:

$$sms(P) = \mu P - r_P P - \delta_P P^2 - \lambda \mu P - G(P)Z \tag{A2}$$

$$sms(Z) = G(P)Z - (\gamma_1 - \gamma_2)G(P)Z - (1 - \gamma_1)G(P)Z - \delta_Z Z^2 \tag{A3}$$

$$sms(D) = \delta_P P^2 + (1 - \gamma_1)G(P)Z + \delta_Z Z^2 - \delta_D D - \frac{\partial}{\partial z}(W_s D) \tag{A4}$$

$$sms(N) = -\mu P + r_P P + \lambda \mu P + (\gamma_1 - \gamma_2)G(P)Z + \delta_D D \tag{A5}$$

where the five terms of the right-hand side in the $sms(P)$ are the phytoplankton growth rate, $\mu$ (is $\mu_{\text{IFL}}(N, I, T)$ or $\mu_{\text{FL}}(N, I, T)$, see Section 2.2), the respiration of $P$, $r_P$ is the respiration rate for $P$ (0.12 day$^{-1}$), the mortality of $P$, $\delta_P$ is the mortality rate for $P$ (0.24 day$^{-1}$), the extracellular excretion of $P$, $\lambda$ is the extracellular excretion rate of $P$ (0.135, no dim.), and the grazing of $P$ by $Z$ ($G(P)$ is the grazing rate equation of Sasai et al., 2016).

$$G(P) = GR_{max}Z(1 - exp(\lambda_P(P2Z - P))) \tag{A6}$$

where $GR_{max}$ is the maximum grazing rate (1.4 day$^{-1}$), $\lambda_P$ is the zooplankton Ivlev constant (1.4 mmol N m$^{-3}$), and $P2Z$ is the zooplankton threshold value for grazing (0.04 mmol N m$^{-3}$). In this study, we adopt two different formulations of phytoplankton growth (InFlexPFT and FlexPFT, see Section 2.2), $\mu$, to the first term of the right-hand side in the $sms(P)$ in Eq. A2. The four terms of $sms(Z)$ are the grazing of $P$ by $Z$, the excretion of $Z$, $\gamma_1$ is the assimilation efficiency of $Z$ (0.7, no dim.), and $\gamma_2$ is the growth efficiency of $Z$ (0.3, no dim.), the egestion of $Z$, and the mortality of $Z$ ($\delta_Z$ is the mortality rate for $Z$ (0.12 day$^{-1}$). The six terms of $sms(D)$ are the mortality of $P$, the egestion of $Z$, the mortality of $Z$, the decomposition from $D$ to $N$, $\delta_D$ is the decomposition rate from $D$ to $N$ (0.3 day$^{-1}$), and the sinking of $D$, $W_s$ is sinking velocity (30 m day$^{-1}$ upper 1000 m or 300 m day$^{-1}$ below 1000 m). The five terms of $sms(N)$ are the phytoplankton growth, the respiration of $P$, the extracellular excretion of $P$, the excretion of $Z$, and the decomposition from $D$ to $N$. More details of biological parameters are described by Sasai et al. (2006, 2010, 2016).

*Author contributions.* YS, SLS, HS and MN designed the study and YS simulated two biological models. SLS also developed the FlexPFT model. ES prepared the satellite and observation data for comparison with the model results. All authors contributed significantly to the writing of the paper.

*Competing interests.* The authors declare that they have no conflict of interest.

*Acknowledgements.* OFES2 simulations were conducted on the Earth Simulator under the support of the Japan Agency for Marine-Earth Science and Technology (JAMSTEC). NASA Goddard Space Flight Center, Ocean Ecology Laboratory, Ocean Biology Processing Group. Moderate-resolution Imaging Spectroradiometer (MODIS) Aqua Chlorophyll Data; 2022 Reprocessing. NASA OB.DAAC, Greenbelt, MD, USA. doi: 10.5067/AQUA/MODIS/L3M/CHL/2022.

*Financial support.* This research has been supported by the Japan Society for the Promotion of Science (JSPS) KAKENHI Grand Numbers, JP17K05663, JP19H05701, JP20K04075, and JRPs-LEAD with DFG.

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

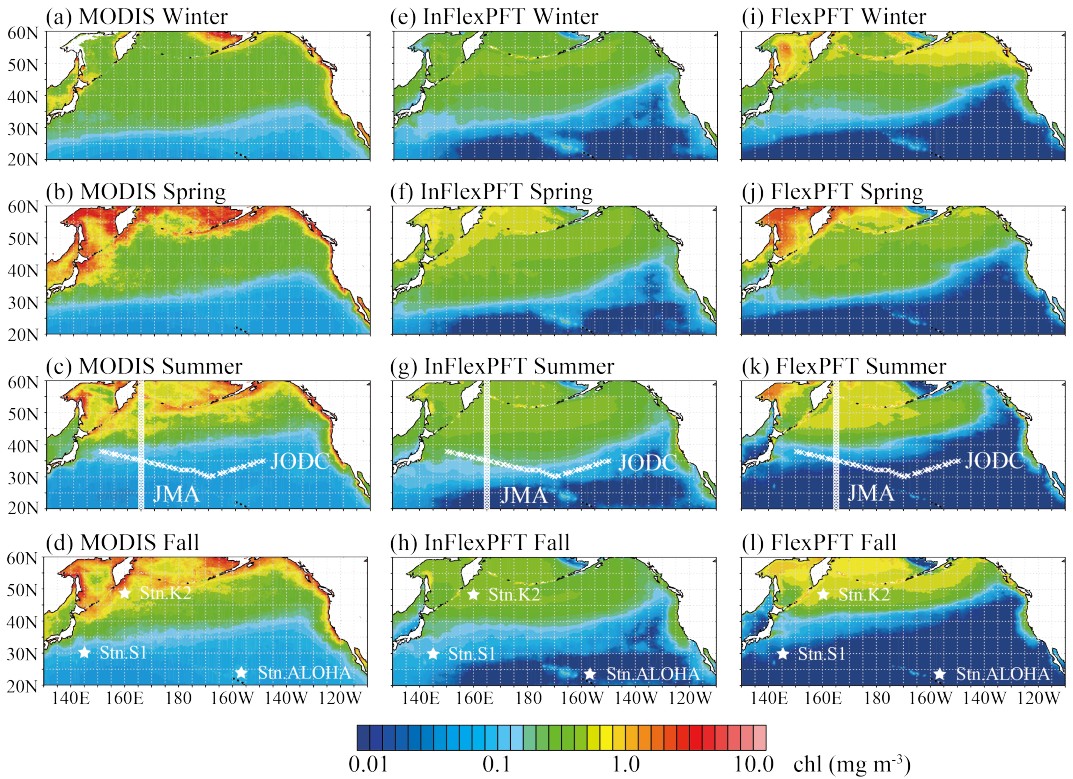

**Figure 1.** Climatological seasonal variations of surface Chl concentration (mg m$^{-3}$) from (a) - (d) MODIS-Aqua imagery and (e) - (i) two models. MODIS-Aqua imagery is averaged from 2003 to 2019 for each season. Two models are averaged from 2000 to 2019 for each season, and show an average of 20 m from the surface layer. White circles and crosses in Figs. 1c, 1g, and 1k show two observation lines (the circles are the locations of the JMA data and the crosses are the locations of JODC data). Three stars in Figs. 1d, 1h, and 1l show the stations of observed time series (Station K2, Station S1, and Station ALOHA).

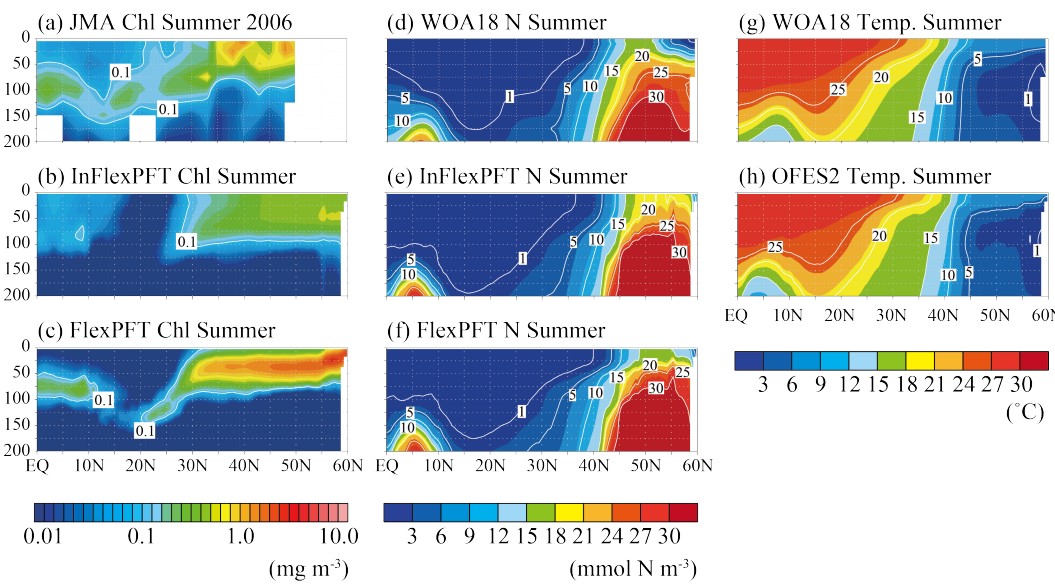

**Figure 2.** Comparison of vertical distributions of (a) - (c) summer (June, July and August) Chl concentration (mg m$^{-3}$), (d) - (f) summer $N$ concentration (mmol N m$^{-3}$), and (g) - (h) summer $T$ ($^\circ$C) along $165^\circ$E (north-south section) in Fig. 1 (white circles). Fig. 2a shows in-situ observation data from the summer of 2006 by the JMA research vessel. Figs. 2d and 2g are from the climatological summer of WOA18. Figs. 2b, 2c, 2e, 2f, and 2h show the models' average value for the summer of 2000 to 2019. White area in Fig. 2a is missing data.

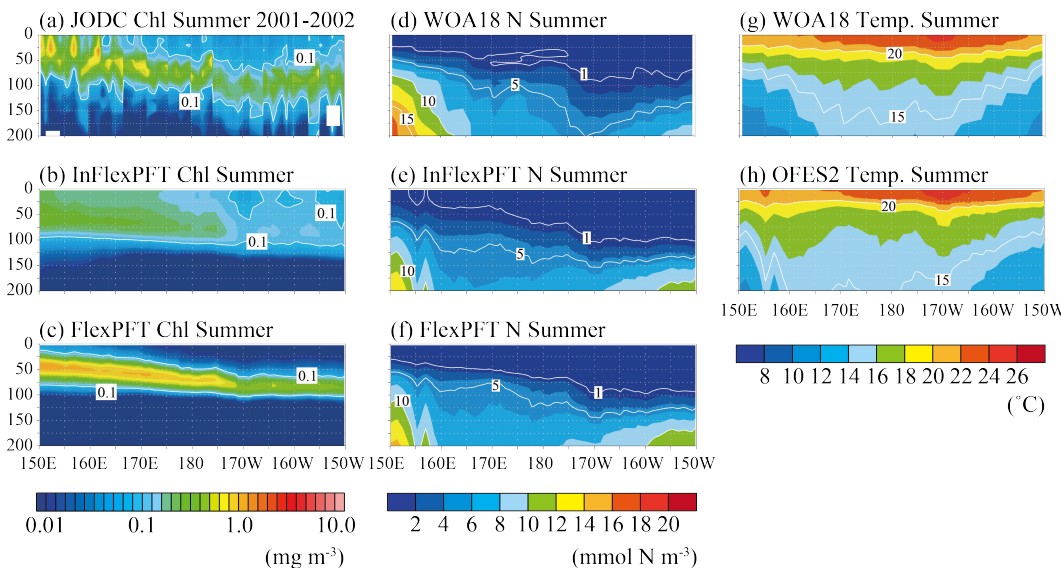

**Figure 3.** Comparison of vertical distributions of (a) - (c) summer (June, July and August) Chl concentration (mg m$^{-3}$), (d) - (f) summer $N$ concentration (mmol N m$^{-3}$), and (g) - (h) summer $T$ (°C) along the location of white crosses (east-west section) in Fig. 1 (white crosses). Fig. 3a shows in-situ observation data collected by the JODC in the summer of 2001-2002. Figs. 3d and 3g are from the climatological summer of WOA18. Figs. 3b, 3c, 3e, 3f, and 3h show the models' average value for the summer of 2000 to 2019. White area in Fig. 3a is missing data.

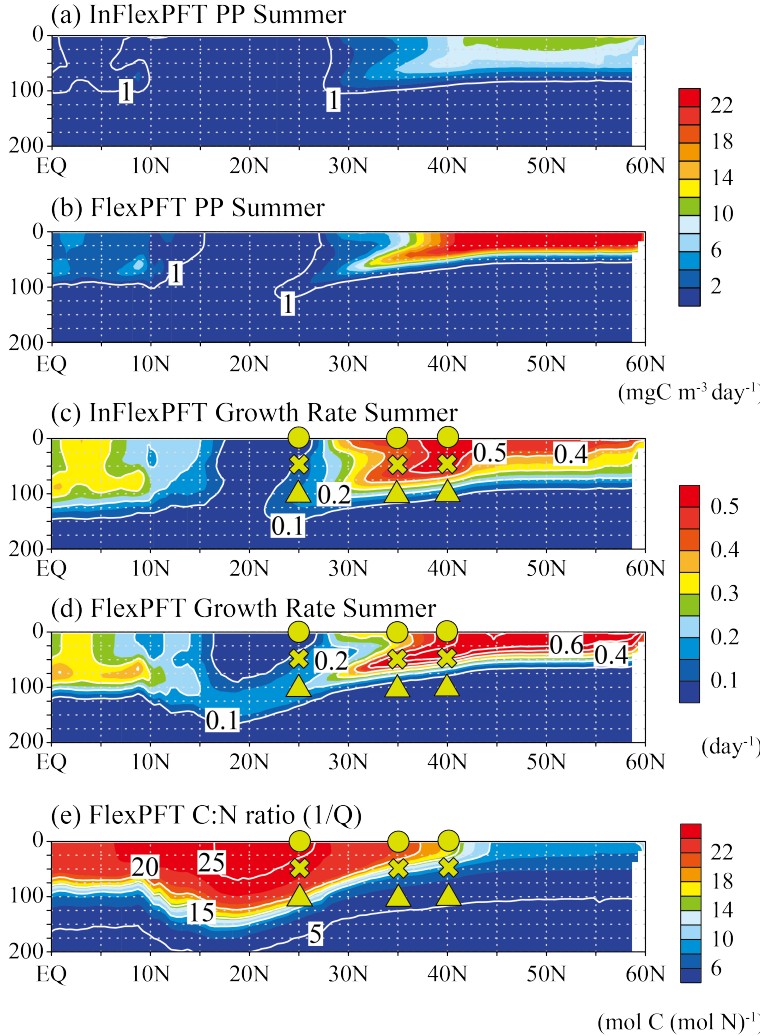

**Figure 4.** Comparison of vertical distributions of (a) - (b) summer primary production (mg C m$^{-3}$ day$^{-1}$), and (c) - (d) summer phytoplankton growth rate (day$^{-1}$) between two models, and (e) vertical distribution of summer C:N ratio in the FlexPFT along 165°E (north-south section) in Fig. 1 (white circles). These figures show the average value for the summer of 2000 to 2019.

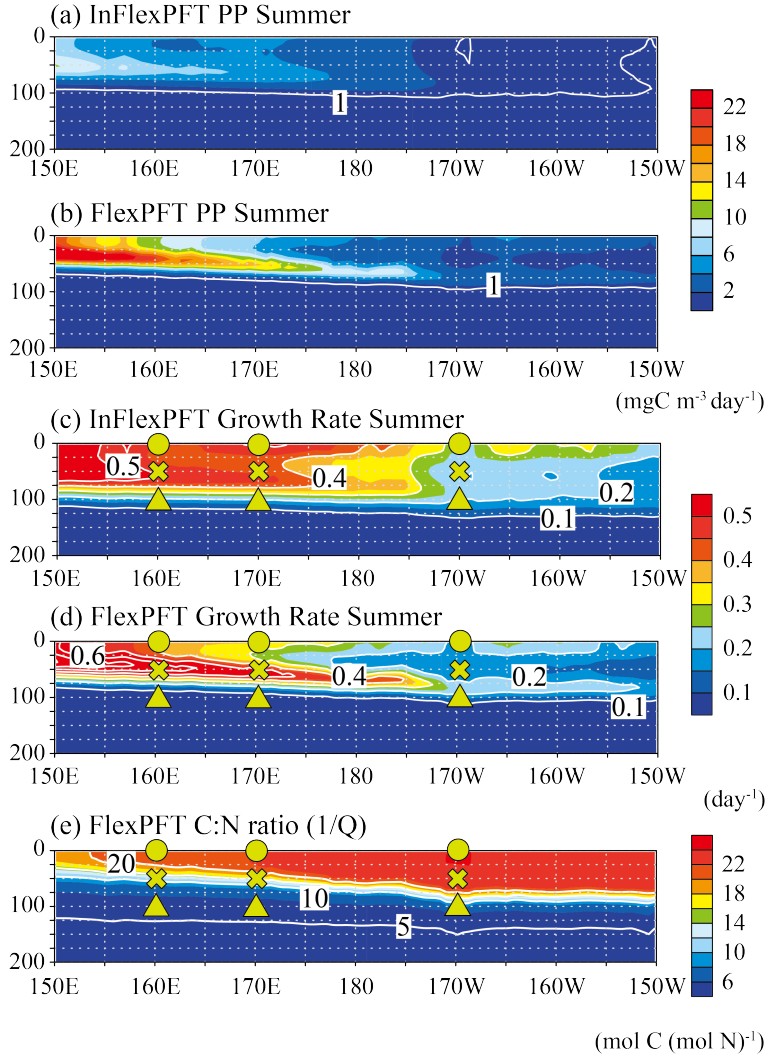

**Figure 5.** Comparison of vertical distributions of (a) - (b) summer primary production (mg C m$^{-3}$ day$^{-1}$), and (c) - (d) summer phytoplankton growth rate (day$^{-1}$) between two models, and (e) vertical distribution of summer C:N ratio in the FlexPFT along the location of white crosses (east-west section) in Fig. 1 (white crosses). These figures show the average value for the summer of 2000 to 2019.

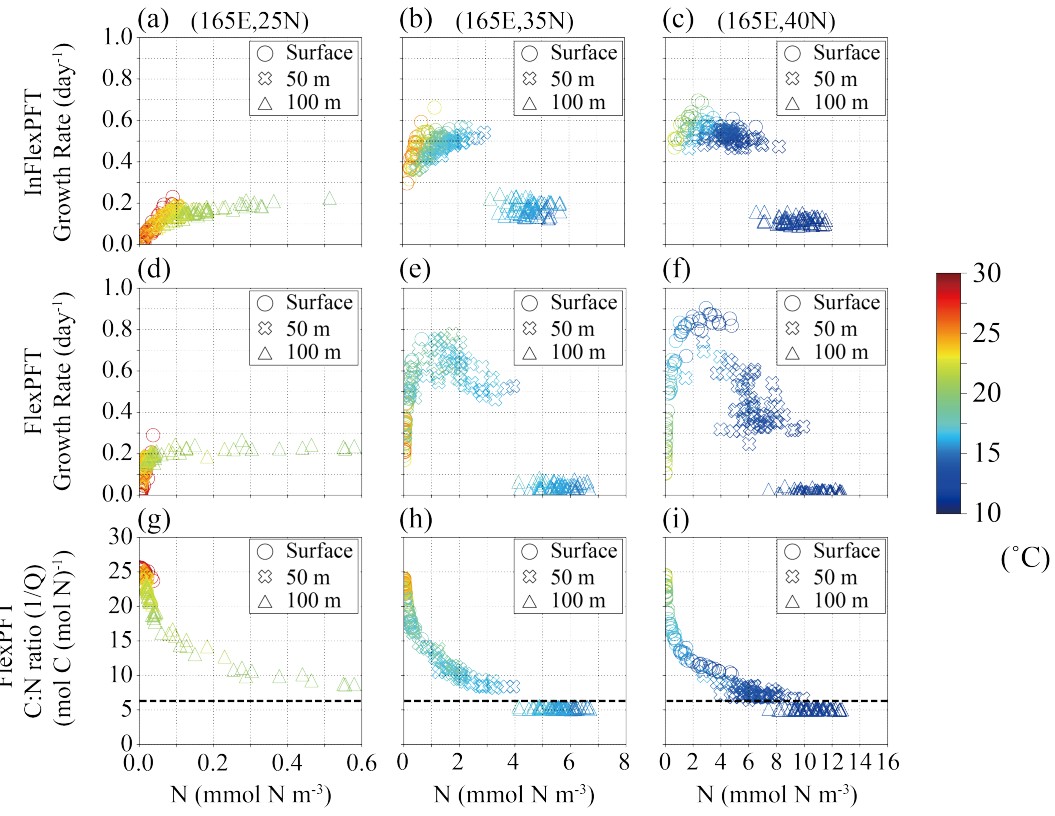

**Figure 6.** Monthly mean (a) - (f) phytoplankton growth rate (day$^{-1}$) in summer of 2000-2019 in Figs. 4c and 4d versus summer monthly mean $N$ concentration (mmol N m$^{-3}$) in Figs. 2e and 2f for each depth (circle is surface, cross is 50 m depth, and triangle is 100 m depth in Figs. 4c and 4d) for each model, and summer monthly mean (g) - (i) C:N ratio versus summer monthly mean $N$ concentration in Flex model. The color for each figure represent the summer monthly mean $T$ (°C) in Fig.2h. Dashed line in Figs. 6g, 6h, and 6i present constant C:N (106:16=6.625) ratio value.

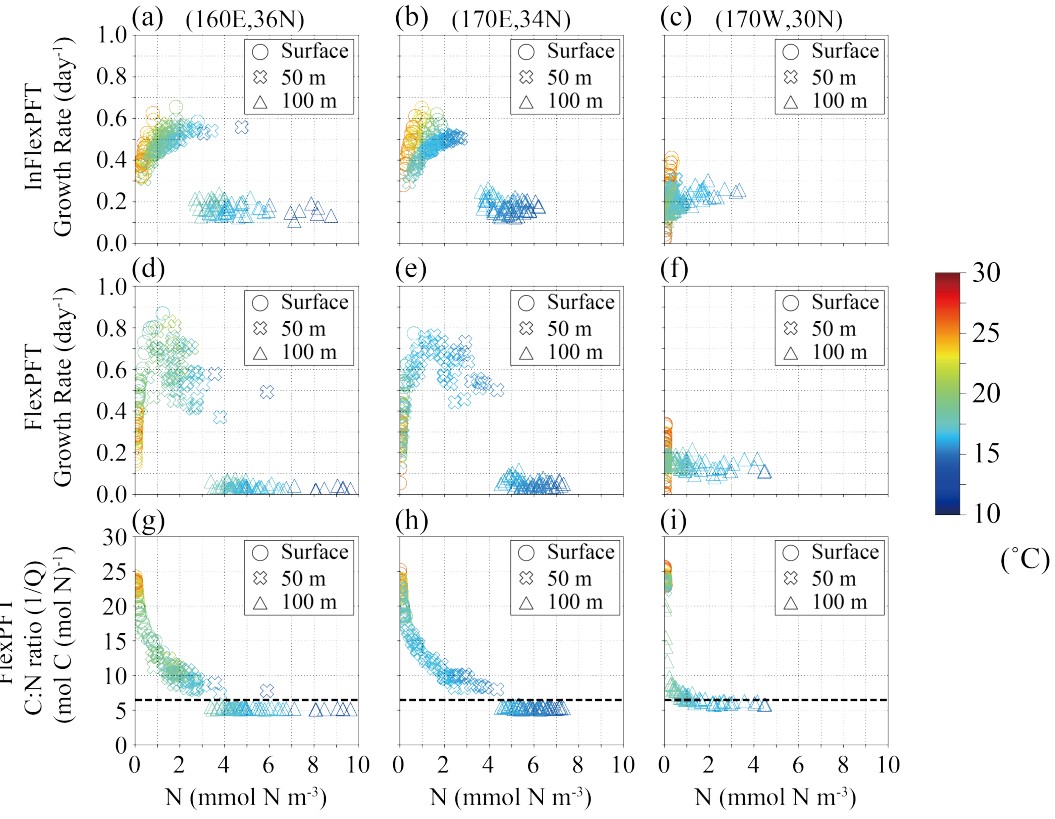

**Figure 7.** Monthly mean (a) - (f) growth rate (day$^{-1}$) in summer of 2000-2019 in Figs. 5c and 5d versus summer monthly mean $N$ concentration (mmol N m$^{-3}$) in Figs. 3e and 3f for each depth (circle is surface, cross is 50 m depth, and triangle is 100 m depth in Figs. 5c and 5d) for each model, and summer monthly mean (g) - (i) C:N ratio versus summer monthly mean $N$ concentration in the FlexPFT model. The color for each figure present summer monthly mean $T$ (°C) in Fig. 3h. Dashed line in Figs. 7g, 7h, and 7i present constant C:N (106:16=6.625) ratio value.

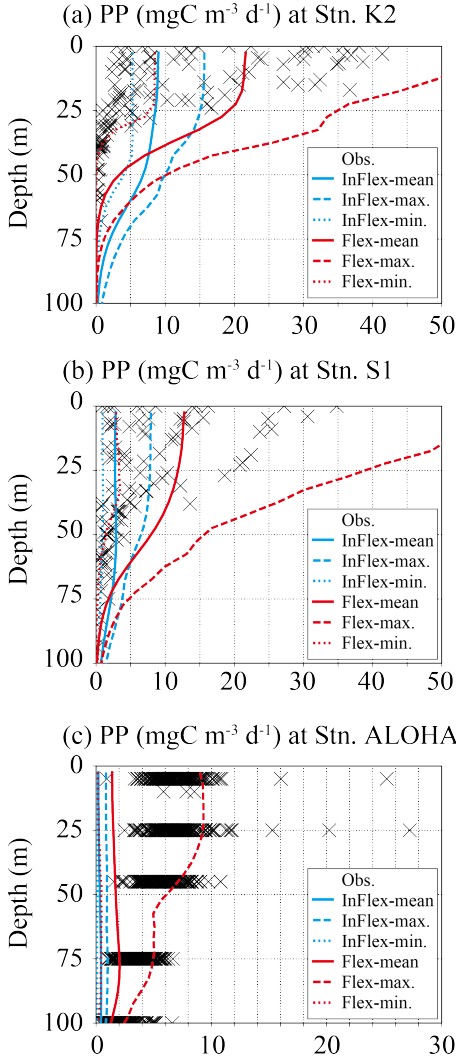

**Figure 8.** Comparison of vertical primary production (mg C m$^{-3}$ day$^{-1}$) between observed station data and two models at (a) Station K2 (47°N,160°E), (b) Station S1 (30°N, 145°E), and (c) Station ALOAH (22.45°N, 158°W) in Fig. 8a. Observed Station K2 data is from 2010 to 2019, Station S1 is from 2010 to 2013, and Station ALOHA is from 2000 to 2019 (crosses). For the two models, the mean primary production of 2000-2019 (solid line), the maximum of 2000-2019 (dashed line), and the minimum of 2000-2019 (dotted line) are shown. Blue lines are InFlexPFT and red lines are FlexPFT. Three stars in Fig. 1 show the observed stations.

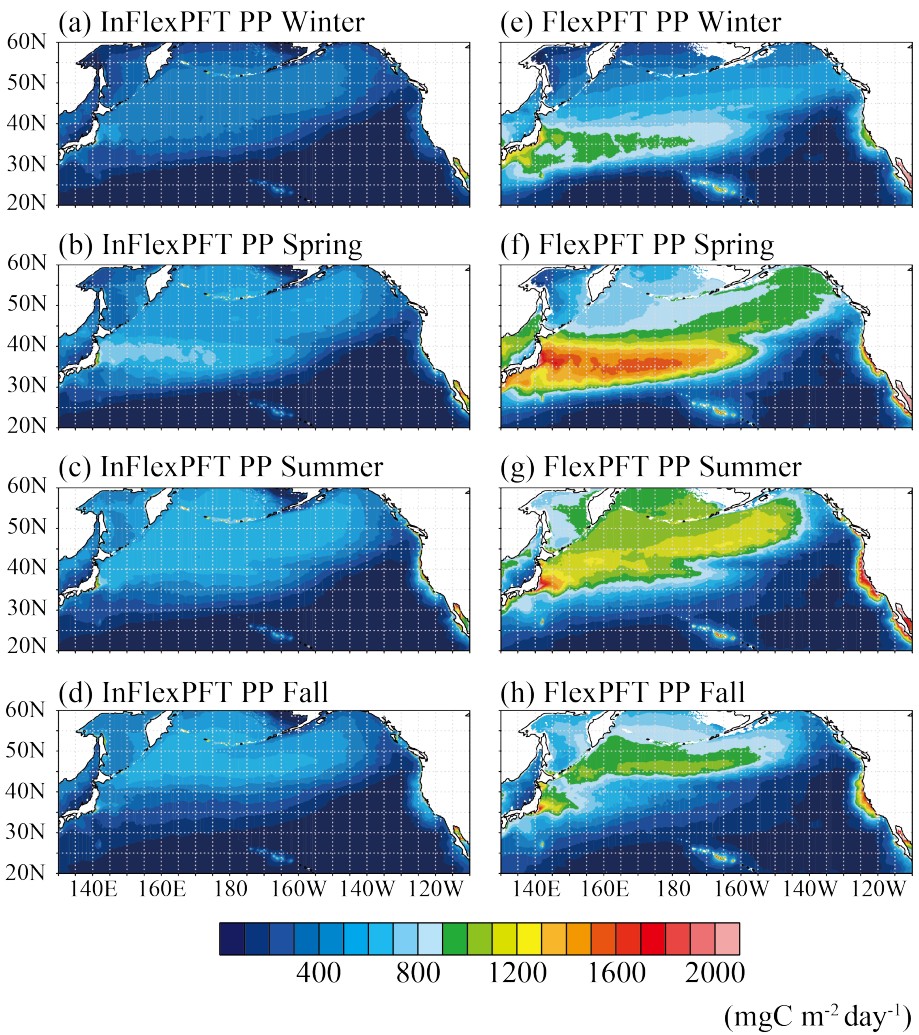

**Figure 9.** Climatological seasonal variations of primary production vertical integrated from surface to 100 m depth (mg C m$^{-3}$ day$^{-1}$) from (a) - (d) the InFlexPFT, and (e) - (h) the FlexPFT. Two models are averaged from 2000 to 2019 for each season.

**Table 1.** Parameter of InFlexPFT and FlexPFT models

| Parameter | Symbol | InFlexPFT | FlexPFT | Unit |
|---|---|---|---|---|
| Potential maximum growth rate | $\mu_{max}$ | 1.5 | 2.2 | $\text{day}^{-1}$ |
| Potential maximum uptake rate for $N$ | $\hat{V}_0$ | 1.0 | 1.0 | $\text{day}^{-1}$ |
| Potential maximum affinity for $N$ | $\hat{A}_0$ | 2.0 | 1.0 | $\text{m}^3 \ (\text{mmol N})^{-1} \ \text{day}^{-1}$ |
| Chl-specific initial slop | $\alpha$ | 2.0 | 2.0 | No dim. |