# Peer review of "Physiological flexibility of phytoplankton impacts modeled chlorophyll and primary production across the North Pacific Ocean"

_EGUsphere, 2022_

## Referee Comment (RC1)

The manuscript *Physiological flexibility of phytoplankton impacts modeled biomass and primary production across the North Pacific Ocean* by Y. Sasai and colleagues assess the importance of optimal nutrient uptake, photoacclimation and variable stoichiometry on the emergence of large scale chlorophyll and primary production patterns in the ocean, using the North Pacific as a testbed. To do that, the authors designed an experiment based on the comparison of two strategic NPZD biogeochemical (BGC) models coupled to a state of the art, eddy-resolving model of ocean physics (OFES2). One of the BGC models implements optimal uptake kinetics (Smith et al 2009), whereas the other incorporates interactive, optimal photoacclimation (Chl:C ratios) and variable stoichiometry (C:N deviating from Redfield ratios) as proposed in the context of the FlexPFT theory of Smith et al 2016). The experiments revealed clear deviations between predictions from both models, with a clear gain in accuracy when using the more complex FlexPFT model (bulk properties like Chl and fluxes like PP closer to observations, better resolution of the deep chlorophyll maximum (DCM), etc.). The authors conclude recommending the adoption of similar approaches by the BGC community.

The topic, approach and results are of great appeal and the manuscript is already an important contribution. The authors designed a clean test of their hypothesis about the emergence of surface and vertical gradients in phytoplankton growth and biomass, and the result clearly support their ideas. However, the manuscript is not easy to follow as it stands, especially the combined *Results and Discussion* section. The narrative in this section is quite descriptive, and it fails to provide a clear picture of the ability of each model to reproduce observed patterns. As commented below, these issues seem to arise in part from the lack of motivation and rationale for model assessment in *Methods*. There are other aspects missing like the limitations of the current model (*e.g.* what about nitrogen fixation?), alternative explanations to the emergence of DCM (Cullen 2015; there may be more recent reviews), or about the performance of similar BGC models in simulating Chl and PP in the Pacific. Together, these issues led me recommend a major revision of the manuscript. I provide some major concerns, and a long list of minor issues and suggestions below. I hope the authors find both lists useful.
* * *
***Major comments***

1. The manuscript is in general well written and structured, but there are two sections that in my opinion deserve another thought, namely *Methods and Materials* and the *Results and Discussion* (see next three points). Although the description of the models is in general easy to follow (see however some minor suggestions below), the fact that the text moves from a complex model to a simpler one is not an optimal choice. I recommend the authors to present first their general approach with the components that are common to both models, and then detail first the simpler model featuring just optimal uptake followed by the more complex model featuring also photoacclimation and variable stoichiometry. I am aware that this might read as a minor issue, but I think it is quite important to ensure that readers realize that, despite their names, FlexPFT is something more complex than InFlexPFT. It is not clear whether one model is a nested version of the other or not (in the sense of a simpler formulation or the result of setting from variables as a constant). For instance, there is certain temptation to just check Table 1 and conclude that InFlexPFT results in reduced Chl and PP when compared with FlexPFT just because $\mu_{\text{max}}$ is lower in the former. There is also

some confusion about whether the model implements only photoacclimation or photoacclimation and variable stoichiometry, and about whether one or both of them are simple NPZDs or not.

2. Merging *Results* and *Discussion* has certain risks. In my opinion, that section of the manuscript needs to dig a bit more into the results and provide more quantitative tests that enable readers to assess the relative merits of each model and to frame the results in the context of similar work. The text reads well, but it lacks any quantitative comparison except toward the end, when the authors comment on the huge variability of available NPP estimates and provide large scale estimates for the overall production of the North Pacific. The manuscript would benefit from a more systematic assessment featuring regional averages (say, at the biome scale?) and some kind of statistical metric or test.

3. There may be other things to say about the choice of the data for the comparison, and about how model output was preprocessed (for instance, how did you process Chl profiles, was there any attempt to mimic the way the ocean color satellites operate?). Since the simulations were forced using JRA-do reanalysis data, one would expect that the target for the models would be to reproduce or match available data. It is not clear what was the aim and objectives of the study, and perhaps that explanation is the only thing missing. The objectives and the rationale for choosing some data and patterns over potential alternatives needs to be justified. The models seems to be doing more than decent job, but the authors need to clarify to what extent some of the apparent biases observed both in surface and subsurface fields reflect are due to biases in simulated physical and chemical conditions or to differences on the phytoplankton model.

4. Finally, a key aspect that the authors need to make clear earlier in the paper is the feasibility that proposed and discussed mechanisms may be actually working in the field. There is room to discuss alternative mechanisms currently ignored by the two models assessed. For instance, interactions between grazers and phytoplankters, potential biases in export and recycling, the metabolic diversity of phytoplankton (*e.g.* nitrogen fixers, picophytoplankton), etc.

5. As a bonus question, although it does not seem central to the study at hand, the formulation of zooplankton grazing was quite intriguing for two reasons the deserve further comment;

    1. the numerical response seems to be nonstandard and deserves further comment, as well as the closure term
    2. Eq (A2) in L429 includes a quadratic mortality term for phytoplankton. That effectively means that phytoplankton dynamics follow logistic growth, which seems redundant with the formulation of phytoplankton growth as a function of available nutrients, and underwater light and temperature conditions, and with the common assumption of a population controlled though grazing by zooplankton. Perhaps I am missing something?
* * *
**Minor comments**
* * *
*Abstract*

L001 - active voice? [Light and nutrient conditions ... ]

L002 - define photoacclimation?

L002 - at the end your model features both photoacclimation and variable stoichiometry, perhaps it is worth highlihgting it

L003 - break the sentence at the comme (it is already a bit twisted), and perhaps join with the next one?

L004 - as commented above, I recommend to go from simple to complex [say optimum nutrient uptake PFT to full FlexPFT], and provide a one sentence description with general details about the two models

L007 - mention OFES2 by name [and acronym]?

L010 - [...] subsurface Chl maxima *in the subtropical gyre* [to provide context]. Otherwise please detail where exactly that happens (especially the overestimation of Chl). As commented above, a figure detailing the magnitude of deviations with satellite data would be very useful.

As a general comment about the abstract; data used for validation is not mentioned at all. Readers might just assume you are testing your model against "oceanic observations" [L003], which may be too vague
* * *
*Introduction*

L035 - not sure if there is something else besides pursuing efficiency and simplicity
[Figure]
 ...

L077 - if InFexlPFT is a typical NPZD, then call it NPZD, or state here too that it is an NPZD implementing optimal uptake kinetics as per Smith et al 2009 [L135ff]?

L080 - perhaps deter giving the full name and details of OFES2 to MatMet?
* * *
*Methods and Materials*

L085 - I understand you extensively modified the simple NPZD to implement either OU kinetics or FlexPFT. I mean, perhaps it is worth mention it and state that the default configuration consisted in a simple NPZD? [or maybe just provide those details later when talking about the BGC component of the model]

L089 - I would put all details about the configuration of the experiments together [L101]. The sentence in L93 is especially intriguing and disconnected from the rest of the explanation [L104].

L113 - Eq (1): perhaps Q(I,T) instead of just Q?

L119 - Eq (2): I really did not like the symbol $\mu_{Flex}(I,T)$, it seems potentially confusing ... what about just $\mu_{\mathrm{max}} , S(I,T) , F(T)$ or $\mu_{Flex}^{*}(I,T)$ [where I would suggest the former] ... otherwise it may lead users to think you need some functional or to iteratively solve the equation?

L124 - Please detail how do you determine maximum affinity [if it is optimized on a daily basis, etc]

L131 - please detail how the optimal value of $\theta$ is updated [I mean, that it is not a constant]

L137 - ideally, it would be nice to see how one can go from Eq (7) to Eq (1) [if that is possible], Otherwise it may be worth stating whether the models are truly nested or they just feature different terms for nutrient the dependent growth [though they propagate to the other terms in FlexPFT]

L153ff - perhaps explicitly include formulas for Chl and PP [$Chl = P , \theta / Q \quad \text{in FlexPFT}$, etc]

L155 - I also miss some details here about what kind of outputs were compared to observations. In principle, since JRA-do is a renaalysis dataset, you might expect a direct match between simulated fields and observations at sea.

L165 - perhaps follow the order physics, chemistry, biology? Again, missing some rationale for the choices and the way the model was evaluated
* * *
*Results and Discussion*

-> *general comment* as an author myself I can understand the preference for pooling the two sections, as a reader I am not such a fan.

L174ff - it seems that the physical component of the model was evaluated elsewhere; if that is the case it would be better to explicitly state so, but it would be ok to go beyond the ability of the model to reproduce major circulation features to mention at least its skill in reproducing temperature and nutrient fields.

L185 - there is certain paradox here since the initial focus of the manuscript on phytoplankton biomass and productivity mutates here on a large section devoted to two sections devoted to chlorophyll (which, needless to say in the context of a photoacclimation paper, is not biomass)

L185 and 200 - perhaps the titles of these sections should somehow detail that they refer to model to data intercomparisons (w.r.t. the section starting on L321)

L190 - please provide some quantitative summary of deviations between models and obs

L200ff - the narrative here becomes a bit difficult to follow to me ... perhaps an alternative structure [grouping results per biome for instance], or just a diagram or table summarizing the main findings [obs, simulated patters, most important regulating factor, etc] might make the section easier to follow

L277 - why? Is it possible to partition the amount of variation due to each to I, N and T?

L327 - horizontally and vertically? Could you develop a bit more what you mean?

L346ff - I think this paragraph belongs to the previous section
* * *
*Conclusions*

L375 - perhaps *acclimation* instead of *adaptive response*?

L409 - perahps hte IA abbreviation can be omitted here to detail instantaneous acclimation

*Appendix A*

L429 - please note comment above about Eq (A2)

L435 - please detail the type of numerical response (*i.e.* no need to force interested readers to go to Sasai et al 2016). My excursion to that paper suggest it is a Gompertz function with a threshold ... did not seem entirely standard (I mean, a commonly used formulation).

*Figures*

Figure 1 - nice maps! Some suggestions doubts to comment in teh text;

- add transect lines to panels c and k?
- what happened in the Gulf of Alaska and at Bering Sea?
- the distinct shape of the gyre suggest there may be underlying biases in ocean physics propagating to chl [what about simulated MLD?]

Figures 2 and 3 - again nice figures and amazing results

- physics, chemistry, biology? [order of columns]
- why not directly comparing data for 2002/2003? [it would be nice to check whether the model reproduces small scale heterogeneity]

Figures 6 and 7

- I like the figures but still feel they fail to clearly convey whether N and I are more important than T ... How would an equivalent figure with T in the abscissa look like? How can you partition which variable contributes more variability?

Figure 8

I think the results for FlexPFT would compare well with satellite based NPP products. Indeed, it would be great if, beyond biases in InFlexPFT the authors can show that actually the simpler model fails to capture large scale gradients [or at least, that is the impression I got].

Figure 9

Is it possible to complement these profiles with a time series plot? [perhaps the monthly climatology at each site]

---

## Author Response (AR1)

(egusphere-2022-91) entitled "Physiological flexibility of phytoplankton impacts modeled biomass and primary production across the North Pacific Ocean" by Sasai et al.

We thank three anonymous reviewers for their valuable comments on our manuscript. Our responses to each reviewers' comment are as follows. (Response to Reviewer #1 is Page 1 to 17. Response to Reviewer #2 is Page 18 to 31. Response to Reviewer #3 is Page 32 to 34.)

Response to Reviewer #1

**The manuscript \*Physiological flexibility of phytoplankton impacts modeled biomass and primary production across the North Pacific Ocean\* by Y. Sasai and colleagues assess the importance of optimal nutrient uptake, photoacclimation and variable stoichiometry on the emergence of large scale chlorophyll and primary production patterns in the ocean, using the North Pacific as a testbed. To do that, the authors designed an experiment based on the comparison of two strategic NPZD biogeochemical (BGC) models coupled to a state of the art, eddy-resolving model of ocean physics (OFES2). One of the BGC models implements optimal uptake kinetics ([Smith et al 2009](https://doi.org/10.3354/meps08022)), whereas the other incorporates interactive, optimal photoacclimation (Chl:C ratios) and variable stoichiometry (C:N deviating from Redfield ratios) as proposed in the context of the FlexPFT theory of [Smith et al 2016](https://doi.org/10.1093/plankt/fbv038)). The experiments revealed clear deviations between predictions from both models, with a clear gain in accuracy when using the more complex FlexPFT model (bulk properties like Chl and fluxes like PP closer to observations, better resolution of the deep chlorophyll maximum (DCM), etc.). The authors conclude recommending the adoption of similar approaches by the BGC community.**

**The topic, approach and results are of great appeal and the manuscript is already an important contribution. The authors designed a clean test of their hypothesis about the emergence of surface and vertical gradients in phytoplankton growth and biomass, and the result clearly support their ideas. However, the manuscript is not easy to follow as it stands, especially the combined \*Results and Discussion\* section. The narrative in this section is quite descriptive, and it fails to provide a clear picture of the ability of each model to reproduce observed patterns. As commented below, these issues seem to arise in part from the lack of motivation and rationale for model assessment in \*Methods\*. There are other aspects missing like the limitations of the current model (\*e.g.\* what about nitrogen fixation?), alternative explanations to the emergence of DCM ([Cullen 2015](https://doi.org/10.1146/annurev-marine-010213-135111); there may be more recent reviews), or about the performance of similar BGC models in simulating Chl and PP in the Pacific. Together, these issues led me recommend a major revision of the manuscript. I provide some major concerns, and a long list of minor issues and suggestions below. I hope the authors find both lists useful.**

We thank Reviewer #1 for valuable comments on our manuscript. The major revised
points are 7 items below. The individual responses (fine characters) to the Reviewer #1's
comments (bold characters) are described after the list of major 7 items.
1.  Title changed "Physiological flexible of phytoplankton impacts modeled chlorophyll
and primary production across the North Pacific". Because phytoplankton biomass
changed chlorophyll.
2.  Introduction
We added the aim and objective in the last paragraph of Introduction.
3.  Methods and Materials
We revised the two subsections "2.2 Formation of Phytoplankton Growth in the
Biological Model" and "2.3 Observed Data" in section 2.
In subsection 2.2, we changed from simple phytoplankton growth rate (InFlexPFT)
description to complex phytoplankton growth rate (FlexPFT) description.
In subsection 2.3, we revised the description of observed data to compare the model
results. The observed year and modeled year are not same, but they are compared to
confirm the reproducibility of the model climatological averaged field (e.g., season).
We revised the first paragraph of subsection 2.3.
4.  Results
We changed section title from "Results and Discussion" to "Results".
We changed three subsection titles.
Title of subsection 3.1 is "Comparison of Surface Chl Pattern".
Title of subsection 3.2 is "Comparison of Vertical Distributions of Chl and PP
along Two transect Lines".
Title of subsection 3.3 is "Vertical Profiles of PP at three stations and PP pattern".
We added the quantitative description in section 3, everywhere.
5.  We added the "Discussion" and revised the "Conclusion".
6.  Figures
Observations points (three stations and two lines) are correctively shown in Figure 1.
Two transect lines (JMA and JODC observation lines) are shown in Figures 1c, 1g,
and 1k in summer map. Three time series stations (Station K2, Station S1, and Station
ALOHA) are shown in Figures 1d, 1h, and 1l in winter map.
The numbers of figures have also been changed because the order of descriptions of
Figures 8 and 9 in the body text has been changed. Figure 8 (old) changes Figure 9
(new), and Figure 9 (old) changes Figure 8 (new).
7.  Appendix A: NPZD model
We added the grazing of phytoplankton by zooplankton equation, G(P), in Appendix
A.
* * *
*Major comments*

1. The manuscript is in general well written and structured, but there are two
sections that in my opinion deserve another thought, namely *Methods and
Materials* and the *Results and Discussion* (see next three points). Although the
description of the models is in general easy to follow (see however some minor
suggestions below), the fact that the text moves from a complex model to a simpler
one is not an optimal choice. I recommend the authors to present first their general
approach with the components that are common to both models, and then detail first
the simpler model featuring just optimal uptake followed by the more complex
model featuring also photoacclimation and variable stoichiometry. I am aware that
this might read as a minor issue, but I think it is quite important to ensure that
readers realize that, despite their names, FlexPFT is something more complex than
InFlexPFT. It is not clear whether one model is a nested version of the other or not
(in the sense of a simpler formulation or the result of setting from variables as a
constant). For instance, there is certain temptation to just check Table 1 and
conclude that InFlexPFT results in reduced Chl and PP when compared with
FlexPFT just because $\mu_{\text{max}}$ is lower in the former. There is also some
confusion about whether the model implements only photoacclimation or
photoacclimation and variable stoichiometry, and about whether one or both of
them are simple NPZDs or not.

Thank you for your comment. Following your suggestion, we have revised the two
sections, "Methods and Materials" and "Results and Discussion". "Methods and
Materials" section was revised including minor comments. "Results and Discussion"
section was divided into "Results" and "Discussion (New section)". Please check the
attached file.

2. Merging *Results* and *Discussion* has certain risks. In my opinion, that section
of the manuscript needs to dig a bit more into the results and provide more
quantitative tests that enable readers to assess the relative merits of each model and
to frame the results in the context of similar work. The text reads well, but it lacks
any quantitative comparison except toward the end, when the authors comment on
the huge variability of available NPP estimates and provide large scale estimates for
the overall production of the North Pacific. The manuscript would benefit from a
more systematic assessment featuring regional averages (say, at the biome scale?)
and some kind of statistical metric or test.

Thank you for your comment. Following your suggestion, we have revised the "Results"
section based on the above and the minor comments below. Please see the attached pdf
file.

3. There may be other things to say about the choice of the data for the comparison,
and about how model output was preprocessed (for instance, how did you process
Chl profiles, was there any attempt to mimic the way the ocean color satellites

**operate?). Since the simulations were forced using JRA-do reanalysis data, one would expect that the target for the models would be to reproduce or match available data.**

Since the 2000s, sea surface chlorophyll data has been accumulated for the last 20 years, and it is possible to analyze the seasonal variability. On the other hand, the vertical profiles of chlorophyll and PP in-situ observed data are mostly snapshots of limited places (time series stations, etc.) and observation lines, and there are few that are spatiotemporally aligned like nutrients and temperature such as WOA database. Since the 2000s, comparable data have been published in the North Pacific and used in model validation. The observational data used for comparison in this study is not sufficient, so it is necessary to prepare publicly available data in order to analyze variations over several days to decade.

New Lines 200-205: We have revised the text to clearly state, "The last 20 years (2000-2019) average of model results were compared with satellite data, in-situ observations, and the climatological data (Chl, nitrate, and temperature). Although the model and observation periods differ somewhat, using the satellite and in-situ observation data observed during the simulation period (2000s), we compare whether the horizontal and vertical patterns of climatological seasonal variations can reproduce the patterns captured by the satellite and the snapshot observations. Especially, we focused on the Chl and PP patterns, which strongly reflect effects of the different assumptions about how growth rates depend on light and nutrients." at the beginning of the Observational Data section.

**It is not clear what was the aim and objectives of the study, and perhaps that explanation is the only thing missing. The objectives and the rationale for choosing some data and patterns over potential alternatives needs to be justified. The models seem to be doing more than decent job, but the authors need to clarify to what extent some of the apparent biases observed both in surface and subsurface fields reflect are due to biases in simulated physical and chemical conditions or to differences on the phytoplankton model.**

New Lines 80-90: We the text to read, "Most of biogeochemical models have similar structure, with nitrogen as the main currency for a simplified food-web, which generally includes phytoplankton and zooplankton, and a regeneration network with detritus, dissolve organic nitrogen, and various nutrients (i.e.,Fasham et al., 1990). Whereas the more complex biogeochemical models have become more common (e.g., Follows et al., 2007, Totterdell, 2019), simple phytoplankton growth (fixed stoichiometry, without photoacclimation) models are still applied widely. In this study, we focus on the acclimative growth response of phytoplankton as incorporated in these models. To evaluate the performance and implications of this acclimative response of phytoplankton growth to varying light and nutrient conditions across the North Pacific Ocean, we compare modeled chlorophyll and primary production from an inflexible phytoplankton control model (InFlexPFT), which assumes fixed C:N:Chl ratios (fixed stoichiometry), to a recently developed phytoplankton model  (FlexPFT, Smith et al., 2016), which incorporates photoacclimation and variable C:N:Chl ratios. We apply these two phytoplankton models in a 3-D eddy-resolving ocean circulation model of the North

Pacific, to assess each model's performance compared to observations of chlorophyll and primary production." In the last paragraph of the Introduction.

**4. Finally, a key aspect that the authors need to make clear earlier in the paper is the feasibility that proposed and discussed mechanisms may be actually working in the field. There is room to discuss alternative mechanisms currently ignored by the two models assessed. For instance, interactions between grazers and phytoplankters, potential biases in export and recycling, the metabolic diversity of phytoplankton (*e.g.* nitrogen fixers, picophytoplankton), etc.**

In this study, we compared the two models, each with only one phytoplankton type: the FlexPFT incorporating variable C:N:Chl ratios and photoacclimation, and the InFlexPFT assuming constant composition without photoacclimation. With the FlexPFT, a single phytoplankton type adjusts its growth rate (i.e., acclimates) depending on available nutrients and light conditions. On the other hand, the InFlexPFT does not account for this physiological flexibility, and therefore either light or nutrient limitation tends to reduce growth rates more with the InFlexPFT compared to the FlexPFT. Also, we ignored other biological processes (e.g., interactions between grazers and phytoplanktons, export and recycling) of BGC model.

New Lines 480-482: We have revised the text to read, "In addition, we will proceed with research on introducing flexible physiology to the growth of multiple phytoplankton, as well as associated food quality effects on predation by zooplankton, and the uncertainty of other biological processes, such as nitrification, grazing, mortality, export and recycling." in the last paragraph of the Conclusions section.

**5. As a bonus question, although it does not seem central to the study at hand, the formulation of zooplankton grazing was quite intriguing for two reasons the deserve further comment;**
**1. the numerical response seems to be nonstandard and deserves further comment, as well as the closure term**

We added zooplankton grazing equation, G(P), and explain this equation in Appendix A.

**2. Eq (A2) in L429 includes a quadratic mortality term for phytoplankton. That effectively means that phytoplankton dynamics follow logistic growth, which seems redundant with the formulation of phytoplankton growth as a function of available nutrients, and underwater light and temperature conditions, and with the common assumption of a population controlled though grazing by zooplankton. Perhaps I am missing something?**

In Eq. (A2), the time-varying formula for P is the growth rate, respiration, mortality, extracellular excretion (a part of growth rate returns N), and grazing by Zooplankton. Since this formula changes only mu contained in the section 2.2, it does not overlap with the growth rate formula pointed out in the comment. Our previous explanation was difficult to understand; therefore, in the revised manuscript we have added a symbol mu_InFlex or mu_Flex after mu in appendix.

----------------------------
**\*Minor comments\***

**\*Abstract\***

**L001 - active voice? [Light and nutrient conditions ... ]**

New Line: We have revised from "…biomass to changing light and nutrient conditions…"
to "… biomass to changes in light and nutrient availability…". Phytoplankton biomass is
passive response.

**L002 - define photoacclimation?**

New Line 2: We have defined "photoacclimation" in the second sentence of the revised
abstract as follows: "…photoacclimation, i.e. the dynamic physiological response of
phytoplankton to varying light and nutrient availability (variable chlorophyll: carbon
ratios)"

**L002 - at the end your model features both photoacclimation and variable**
**stoichiometry, perhaps it is worth highlighting it**

New Line 2: Yes, it is highlighting message in our manuscript. As mentioned in the
comment above, we have added the definition of "photoacclimation", and we now state
clearly that the FlexPFT accounts for both photoacclimation and variable composition.

**L003 - break the sentence at the comme (it is already a bit twisted), and perhaps**
**join with the next one?**

New Lines 4-5: We revised from "… their application and testing against oceanic
observations remain limited." to "… their application and testing against the observed
flexible response of phytoplankton communities remains limited.".

**L004 - as commented above, I recommend to go from simple to complex [say**
**optimum nutrient uptake PFT to full FlexPFT], and provide a one sentence**
**description with general details about the two models**

New Lines 6-9: As mentioned in the comment, we revised "We compare modeled
chlorophyll and primary production from an inflexible control model (InFlexPFT), which
assumes fixed carbon (C):nitrogen (N):chlorophyll (Chl) ratios, to a recently developed
flexible phytoplankton functional type model (FlexPFT), which incorporates
photoacclimation and variable C:N:Chl ratios.".

**L007 - mention OFES2 by name [and acronym]?**

OFES2 is not mentioned in abstract, but it is described in the body text (Section 2).

**L010 - [...] subsurface Chl maxima \*in the subtropical gyre\* [to provide context]. Otherwise please detail where exactly that happens (especially the overestimation of Chl). As commented above, a figure detailing the magnitude of deviations with satellite data would be very useful.**

New Line 12: We add "in the subtropical gyre" after "subsurface Chl maxima".

**As a general comment about the abstract; data used for validation is not mentioned at all. Readers might just assume you are testing your model against "oceanic observations" [L003], which may be too vague**

New Lines 10: We added the "(e.g., satellite imagery and vertical profiles of in-situ observations)" after "We coupled each phytoplankton model … and evaluate their respective performance versus observations". As the [L003] comment, we revised it. The term "oceanic observations" was ambiguous, so we revised it to "observed flexible response of phytoplankton communities".
* * *
**\*Introduction\***

**L035 - not sure if there is something else besides pursuing efficiency and simplicity ;) ...**

Yes, I removed "for the sake of computational efficiency and simplicity.".

**L077 - if InFexlPFT is a typical NPZD, then call it NPZD, or state here too that it is an NPZD implementing optimal uptake kinetics as per Smith et al 2009 [L135ff]?**

New Lines 84-90: As the [L077] and [L080] comments, we revised it as follows:
"In this study, we focus on the acclimative growth response of phytoplankton as incorporated in these models. To evaluate the performance and implications of this acclimative response of phytoplankton growth to varying light and nutrient conditions across the North Pacific Ocean, we compare modeled chlorophyll and primary production from an inflexible phytoplankton control model (InFlexPFT), which assumes fixed C:N:Chl ratios (fixed stoichiometry), to a recently developed phytoplankton model (FlexPFT, Smith et al., 2016), which incorporates photoacclimation and variable C:N:Chl ratios. We apply these two phytoplankton models in a 3-D eddy-resolving ocean circulation model of the North Pacific, to assess each model's performance compared to observations of chlorophyll and primary production."

**L080 - perhaps deter giving the full name and details of OFES2 to MatMet?**

We moved the sentence of full name and details of OFES2 to "Methods and Materials". We revised from "… coupled physical-biological model of the North Pacific, consisting of the OFES2 including…" to "… coupled physical-biological model of the North Pacific, consisting of the physical ocean model (OFES2, namely the Ocean general circulation model For the Earth Simulator) coupled with a simple nitrogen based Nitrate-Phytoplankton-Zooplankton-Detritus (NPZD)…" in the first paragraph of the Section 2.1.
* * *
**\*Methods and Materials\***

**L085 - I understand you extensively modified the simple NPZD to implement either OU kinetics or FlexPFT. I mean, perhaps it is worth mention it and state that the default configuration consisted in a simple NPZD? [or maybe just provide those details later when talking about the BGC component of the model]**

In this study, we only changed the term of phytoplankton growth rate in the simple NPZD model. I don't think it is necessary for readers who understand the BGC model, but readers who are not very familiar with the BGC model or who will conduct research to discuss the uncertainty of phytoplankton growth rates and other biological processes (e.g., grazing, mortality, export and recycling, and nitrogen fixation) in the future. I think a short NPZD model description would be helpful for them.

**L089 - I would put all details about the configuration of the experiments together [L101]. The sentence in L93 is especially intriguing and disconnected from the rest of the explanation [L104].**

New Lines 103-104: We revised from "The last day of 1979 is used for the initial physical fields for this simulation." to "The last day of 1979 is used for the initial physical fields for performing coupled physical-biological model simulation." in [L93].

New Lines 118-119: We revised from "Two NPZD models are incorporated after the last day of 1979 of the OFES2." to "Two NPZD models are incorporated after the last day of 1979 of the physical fields in the OFES2" in [L104].

**L113 - Eq (1): perhaps Q(I,T) instead of just Q?**

In Section 2.2. Yes, you are right. Q and fv are functions of I, T, and N. In the explanation of the formula, we changed Q and fv to Q(N,I,T) and fv(N,I,T).

**L119 - Eq (2): I really did not like the symbol $\mu_{Flex}(I,T)$, it seems potentially confusing ... what about just $\mu_{\mathrm{max}} \, S(I,T) \, F(T)$ or $\mu_{Flex}^{*}(I,T)$ [where I would suggest the former] ... otherwise it may lead users to think you need some functional or to iteratively solve the equation?**

Thank you for your suggestion. We changed the symbol "mu_{Flex}(I,T)" to "mu_{max} S(I,T) F(T)" in Eqs. 5 and 7 in Section 2.2.

**L124 - Please detail how do you determine maximum affinity [if it is optimized on a daily basis, etc]**

New Lines 178-180: These parameter values were determined by tuning the model to reproduce the seasonal and spatial variability of N and Chl in the near-surface of the North Pacific. We revised "Parameter values, mu_max, V_0, A_0, and alpha (Table 1) used in Eqs. 1 to 7 for the phytoplankton growth rate were tuned, separately for each coupled model, to reproduce the seasonal variability of N, and Chl in the near-surface of North Pacific.". after explanation of equations.

**L131 - please detail how the optimal value of $\theta$ is updated [I mean, that it is not a constant]**

Formula theta is so complicated that we will not go into details here, but just cite two papers (Pahlow et al., 2013 and Smith et al., 2016). The optimal value of theta is calculated and applied only when irradiance I is greater than the threshold irradiance; otherwise, when light levels are insufficient to justify the respiratory cost of chlorophyll synthesis, the model assumed that no new chlorophyll is produced. In the latter case, theta is set to a constant value (no photoacclimation).

New Lines 174-176: We have revised by adding: "The optimal value of Chl:C ratio in the FlexPFT is applied when irradiance I exceeds the threshold irradiance, below which the respiratory cost outweighs the benefits of producing chlorophyll (Pahlow et al., 2013 and Smith et al., 2016)."

**L137 - ideally, it would be nice to see how one can go from Eq (7) to Eq (1) [if that is possible], Otherwise it may be worth stating whether the models are truly nested or they just feature different terms for nutrient the dependent growth [though they propagate to the other terms in FlexPFT]**

In the InFlexPFT, N-limitation and light-limitation have independent (multiplicative) effects on growth. In the FlexPFT, the trade-off between light- and nutrient- acquisition is built into the formulations, resulting in inter-dependent effecs. Therefore, there is not clear way to migrate simply from the InFlexPFT growth equation to the FlexPFT growth equation. Here, we will keep the difference in the formula of growth and leave any such derivation for future work.

**L153ff - perhaps explicitly include formulas for Chl and PP [$Chl = P \, \theta / Q \quad \text{in FlexPFT}$, etc]**

New Lines 193-198: We have added "Chl = P x theta/Q" and "PP = mu_Flex x P x 1/Q". etc.

**L155 - I also miss some details here about what kind of outputs were compared to observations. In principle, since JRA-do is a reanalysis dataset, you might expect a direct match between simulated fields and observations at sea.**

Many studies of physical fields using OFES2 output have been carried out. In particular, the reproducibility of OFES2 has been reported in Sasaki et al. (2020) and others. Here, we want to discuss the difference in phytoplankton biomass and production depending on the growth formula, so we keep it to the minimum verification for physical fields (temperature and nutrients).

**L165 - perhaps follow the order physics, chemistry, biology? Again, missing some rationale for the choices and the way the model was evaluated**

Verification of model results requires data (physical, biogeochemical fields) on various spatiotemporal scales. Verification of modeled temperature and nutrient distribution, which are functions of growth, is carried out only in comparable cross-sections (2 lines). In this study, we discuss seasonal variability of climatological values in the biological fields as an example. Impacts not considered this time (circulation, mixing, etc.) will be discussed in the future.
* * *
**\*Results and Discussion\***

**-> \*general comment\* as an author myself I can understand the preference for pooling the two sections, as a reader I am not such a fan.**

We divided this section into "Results" and "Discussion".

**L174ff - it seems that the physical component of the model was evaluated elsewhere; if that is the case it would be better to explicitly state so, but it would be ok to go beyond the ability of the model to reproduce major circulation features to mention at least its skill in reproducing temperature and nutrient fields.**

New Lines 225-226: We added "In addition, the seasonal variability of T and N fields in the near-surface over the North Pacific are also well reproduced (not shown)." between "The eddy-resolving ocean … mesoscale eddies, and upwelling events." and "These physical processes …".

**L185 - there is certain paradox here since the initial focus of the manuscript on phytoplankton biomass and productivity mutates here on a large section devoted to two sections devoted to chlorophyll (which, needless to say in the context of a photoacclimation paper, is not biomass)**

New Lines 229-238: We revised the description of results section in the first paragraph before the 3.1 section.
From "Here we focus on the different assumptions about how phytoplankton growth rate depends on ambient nitrogen concentration and light intensity. First, the reproducibility of seasonal and horizontal Chl distributions is described. Next, we compare the results of the two coupled physical-biological models in terms of Chl and PP along two vertical transects (north-south and east-west, respectively) in the North Pacific, and discuss the reasons for the differences. Finally, the difference in PP as calculated by these two models over the North Pacific is also discussed."

to "Here we focus on the different assumptions about how phytoplankton growth rate depends on ambient nitrogen concentration and light intensity. First, the reproducibility of seasonal and horizontal Chl distributions is described. As the Chl concentration in the FlexPFT is calculated from P x theta /Q, and reflects the changes in theta and Q, we examine how variations in C:N:Chl ratios impact the surface Chl pattern. Next, we compare the results of the two coupled physical-biological models in terms of Chl and PP along two vertical transects (north-south and east-west, respectively) in the North Pacific, and discuss the reasons for the differences. Especially, the role of photoacclimation in the formation of SCM and the growth rate on the variable C:N:Chl ratios of phytoplankton. Finally, the difference in PP as calculated by these two models over the North Pacific and the comparison with limited PP vertical profiles are discussed. The extent to which the different growth rate (InFlexPFT vs FlexPFT) affects the estimated PP is described.".

**L185 and 200 - perhaps the titles of these sections should somehow detail that they refer to model to data intercomparisons (w.r.t. the section starting on L321)**

We changed two section's titles.
1. New Line 261: "3.2 Comparison of Vertical Distributions of Chl and PP along the Two Transects Lines"
2. New Line 386: "3.3 Vertical Profiles of PP at Three Stations and PP Patterns" .

**L190 - please provide some quantitative summary of deviations between models and obs**

New Line: We added quantitative discussion in "Results" section (see attached pdf file).

**L200ff - the narrative here becomes a bit difficult to follow to me ... perhaps an alternative structure [grouping results per biome for instance], or just a diagram or table summarizing the main findings [obs, simulated patters, most important regulating factor, etc] might make the section easier to follow**

We revised the "Results" section. See attached file.

**L277 - why? Is it possible to partition the amount of variation due to each to I, N and T?**

It is possible to separate and present the limitation factors. However, here we examine the differences by simultaneously plotting the three factors (I is different symbol, N is horizontal axis, and T is color) that control the growth rate. In addition, we explain the effect of the C: N ratio in the FlexPFT.

**L327 - horizontally and vertically? Could you develop a bit more what you mean?**

New Lines 408-410: We added "At the gyre boundary, in addition to the surface, primary production is greater compared to other regions. Because the nutricline depth (close to the base of the euphotic layer) and the light intensity are optimal for the spring production." after "… the spring bloom occurs both horizontally and vertically.".

**L346ff - I think this paragraph belongs to the previous section**

New Lines 386-426: We changed the order of this section and the previous section.
Old Figure numbers 8 and 9 changed to new figure numbers 9 and 8.
* * *
*Conclusions*

**L375 - perhaps *acclimation* instead of *adaptive response*?**

New Line 469: We revised from "adaptive response" to "acclimation".

**L409 - perhaps the IA abbreviation can be omitted here to detail instantaneous acclimation**

We deleted "IA" as the comment.
* * *
*Appendix A*

**L429 - please note comment above about Eq (A2)**

Same response to Major comment 5.

**L435 - please detail the type of numerical response (*i.e.* no need to force interested readers to go to Sasai et al 2016). My excursion to that paper suggest it is a Gompertz function with a threshold ... did not seem entirely standard (I mean, a commonly used formulation).**

We added G(P) equation (A6) after A5.
* * *
*Figures*

**Figure 1 - nice maps! Some suggestions doubts to comment in the text;**

**- add transect lines to panels c and k?**

We added transect lines to Figs.1 c and k.

**- what happened in the Gulf of Alaska and at Bering Sea?**

In these regions (low Chl), the modeled nitrate concentration in the surface layer is depleted (not shown). Possibility, in the shallow Bering Sea (East side), nutrient supply is little by the modeled physical processes (e.g., circulation or tidal mixing between marginal sea and open ocean) or is no river inflow (not include). Therefore, the physical
model needs to be improved.
**- the distinct shape of the gyre suggests there may be underlying biases in ocean**
**physics propagating to chl [what about simulated MLD?]**
The model MLD reproduced its seasonal variations. Comparing OFES2 with WOA, the
climatological MLD in the model tends to be deeper (25 – 50 m in March) in the boundary
of two gyres (not shown). In this study, we are comparing by climatological value, so we
would like to discuss the Chl biases due to the mixed layer in future.
**Figures 2 and 3 - again nice figures and amazing results**
**- physics, chemistry, biology? [order of columns]**
It is shown in order to emphasize the difference in reproducibility of the vertical
distribution of chlorophyll. The reproducibility of temperature and nutrient distributions
depends on the physical fields of the model. Temperature and nutrient environments are
important for biomass such as chlorophyll, but the formula is controlled by BGC model,
and even if the temperature and nutrient environment is the same, the bias becomes large.
**- why not directly comparing data for 2002/2003? [it would be nice to check whether**
**the model reproduces small scale heterogeneity]**
In the data of JODC, the observed location, depth, and time are different. Perhaps it is
capturing variations of chlorophyll on small scale. We have checked the similar
patchiness of daily mean chlorophyll (Simulated date is July, $1^{st}$, 2001) pattern in the
model along the same transect line (See attached file). Its chlorophyll pattern in the
FlexPFT model reflects the effect of small variability of vertical nitrate pattern (1 mmol
N $m^{-3}$). Even in the InFlexPFT model, a smaller scale pattern can be reproduced, but the
SCM is not clear. This is future study to investigate the impact of smaller scale process
on the chlorophyll distribution.

[Figure]

**Figures 6 and 7**

**- I like the figures but still feel they fail to clearly convey whether N and I are more important than T ... How would an equivalent figure with T in the abscissa look like? How can you partition which variable contributes more variability?**

T-limitation is also important around 20 degree C, which is reference temperature in T-limitation equation, (in the subtropical region) for growth rate. Especially, in the subsurface layer (50m in below figure), Figures 6e and 7e show the higher growth rate compared with the that of surface layer (same N concentration and strong light intensity). Figure (e) (165E, 35N), where is the boundary between two gyres, shows the high growth rate around 20 degree C in the subsurface layer (50m). At other locations, the effect of T-limitation for growth rate is smaller than the N- and light-limitations.

[Figure]

In the east-west line, the similar pattern shows below. As the locations in Figures (d) and
(e) are close to (165E, 35N), and the FlexPFT growth rate is the highest around 20 degree
C.

[Figure]

**Figure 8**

**I think the results for FlexPFT would compare well with satellite based NPP**
**products. Indeed, it would be great if, beyond biases in InFlexPFT the authors can**
**show that actually the simpler model fails to capture large scale gradients [or at least,**
**that is the impression I got].**

Satellite based NPP has a large range depending by the formula of NPP to estimate (e.g.,
Kulk et al., 2020). Attached figure (below) shows the seasonal variability of NPP
distribution map with three different (CAFE, CBPM, and VGPM) estimates using
satellite data for comparison of modeled estimation. CAFE is the Carbon, Absorption, and Fluorescence Euphotic-resolving net primary production model, which is an
adaptable framework for advancing global ocean productivity assessments by exploiting
state-of-the-art satellite ocean color analyses and addressing key physiological and
ecological attributes of phytoplankton (Silsbe et al., 2016,
https://doi.org/10.1002/2016GB005521). CbPM is the Carbon-based Production Model,
where inorganic carbon is fixed by photosynthetic processes (Behrenfeld et al., 2005,
https://doi.org/10.1029/2004GB002299). VGPM is the Vertically Generalized
Production Model, which is a chlorophyll-based model that estimate net primary
production from chlorophyll, available light, and the photosynthetic efficiency
(Behrenfeld and Falkowski 1997). These data are from Ocean Productivity web site
(https://sites.science.oregonstate.edu/ocean.productivity/).
Due to the large variability of three satellite based NPP map, only the differences
between the models are shown in this study.

[Figure]

[Figure]

(a) InFlexPFT PP Winter (e) FlexPFT PP Winter (b) InFlexPFT PP Spring (f) FlexPFT PP Spring (c) InFlexPFT PP Summer (g) FlexPFT PP Summer (d) InFlexPFT PP Fall (h) FlexPFT PP Fall

$(mgC\ m^{-2}\ day^{-1})$

**Figure 9**

**Is it possible to complement these profiles with a time series plot? [perhaps the monthly climatology at each site]**

New Line 387: Modeled data shows the daily mean during 2000 to 2019. We added "modeled daily mean" in the first sentence of section 3.3.

Response to Reviewer #2

**General Comments**
**The manuscript "Physiological flexibility of phytoplankton impacts modeled biomass and primary production across the North Pacific Ocean" by Y. Sasai and colleagues compares modeled phytoplankton biomass and primary production from a flexible plankton community model accounting for photoacclimation and variable C:N:Chl, with an inflexible plankton community model assuming constant C:N:Chl ratios. These models are coupled to a 3-D eddy-resolving ocean circulation model of the North Pacific. The authors compare the performance of these models by using Chl, nutrient, and primary production observations and find that primary production and chlorophyll were better predicted/modeled by incorporating photoacclimation and variable C:N:Chl ratios.**
**This manuscript provides valuable results that are important for the future implementation of plankton community models. However, as the manuscript stands, I suggest major revisions to outlay a more clear motivation and revise the methods, results, and discussion sections to allow readers to more easily follow this manuscript.**

We thank Reviewer #2 for valuable comments on our manuscript. The major revised points are 5 items below. The individual responses (fine characters) to the Reviewer #2's comments (bold characters) are described after the list of major 5 items.

8.  Title changed "Physiological flexible of phytoplankton impacts modeled chlorophyll and primary production across the North Pacific". Because phytoplankton biomass changed chlorophyll.

9.  Introduction
    We added the aim and objective in the last paragraph of Introduction.

10. Methods and Materials
    We revised the two subsections "2.2 Formation of Phytoplankton Growth in the Biological Model" and "2.3 Observed Data" in section 2.

    In subsection 2.2, we changed from simple phytoplankton growth rate (InFlexPFT) description to complex phytoplankton growth rate (FlexPFT) description.

    In subsection 2.3, we revised the description of observed data to compare the model results. The observed year and modeled year are not same, but they are compared to confirm the reproducibility of the model climatological averaged field (e.g., season). We revised the first paragraph of subsection 2.3.

11. Results
    We changed section title from "Results and Discussion" to "Results".
    We changed three subsection titles.
        Title of subsection 3.1 is "Comparison of Surface Chl Pattern".

Title of subsection 3.2 is "Comparison of Vertical Distributions of Chl and PP along Two transect Lines".

Title of subsection 3.3 is "Vertical Profiles of PP at three stations and PP pattern". We added the quantitative description in section 3, everywhere.

12. We added the "Discussion" and revised the "Conclusion".
* * *
**Specific comments:**

**1.       There should be a more clearly description of the structural differences between models. Although the description of the models is easy to follow, there is some confusion about what the key differences between models are. For example, throughout the manuscript, the text deviates on whether only the complex model implements photoacclimation or both models do. In Table 1, the differences in potential maximum growth rates can create confusion on whether it is the same model simply having a higher growth rate, or understanding where the main differences between models are coming from.**

Thank you for your comment. As your suggestion, we revised the "Methods and Materials" section based on the suggestions above and the minor comments below. In the revised manuscript, we now specify that only the FlexPFT model includes photoacclimation.    Please see the attached manuscript file.

Description of difference in maximum growth rate was added in section 2.2.
New Lines 183-186: "For example, the potential maximum growth rate, mu_max, is 1.5 (day^-1) for the InFlexPFT, compared to 2.2 (day^-1) for the FlexPFT. Increasing the potential maximum growth rate decreases the surface $N$ concentration in the subpolar gyre, to the point of depleting nutrient during summer, while increasing the surface Chl concentration across the whole gyre."

**1.       The results section can be hard to follow in some parts, and quantitative information backing up the results stated will allow readers to better understand the variation between models and models and observations.**

Thank you for your comment. Following your suggestion, we revised the "Results" section based on this comment and other minor comments. Please see the attached file.

**1.       The aims and objectives of the study are lacking throughout the manuscript, especially when stating what observations are being used.**

New Lines 80-90: We the text to read, "Most of biogeochemical models have similar structure, with nitrogen as the main currency for a simplified food-web, which generally includes phytoplankton and zooplankton, and a regeneration network with detritus, dissolve organic nitrogen, and various nutrients (i.e.,Fasham et al., 1990). Whereas the more complex biogeochemical models have become more common (e.g., Follows et al., 2007, Totterdell, 2019), simple phytoplankton growth (fixed stoichiometry, without photoacclimation) models are still applied widely. In this study, we focus on the
acclimative growth response of phytoplankton as incorporated in these models. To
evaluate the performance and implications of this acclimative response of
phytoplankton growth to varying light and nutrient conditions across the North Pacific
Ocean, we compare modeled chlorophyll and primary production from an inflexible
phytoplankton control model (InFlexPFT), which assumes fixed C:N:Chl ratios (fixed
stoichiometry), to a recently developed phytoplankton model   (FlexPFT, Smith et al.,
2016), which incorporates photoacclimation and variable C:N:Chl ratios. We apply
these two phytoplankton models in a 3-D eddy-resolving ocean circulation model of the
North Pacific, to assess each model's performance compared to observations of
chlorophyll and primary production." In the last paragraph of the Introduction.
**There needs to be a better explanation of why this data was used, and why**
**comparing the last 20 years of the model run with observations from different**
**years instead of exact comparisons?**
Since the 2000s, sea surface chlorophyll data has been accumulated for the last 20
years, and it is possible to analyze the seasonal variability. On the other hand, the
vertical profiles of chlorophyll and PP from in-situ observations are mostly snapshots of
limited places (time series stations, etc.) and observation lines, and there are few that are
spatiotemporally aligned with nutrients and temperature, as in the WOA database. Since
the 2000s, comparable data have been published in the North Pacific and used for model
validation. For a more through comparison and assessment of model performance, it
will be necessary to prepare more publicly available data in order to analyze variations
over different timescales, from days to decades.
New Lines 200-205: We have revised the text to clearly state, "The last 20 years (2000-
2019) average of model results were compared with satellite data, in-situ observations,
and the climatological data (Chl, nitrate, and temperature). Although the model and
observation periods differ somewhat, using the satellite and in-situ observation data
observed during the simulation period (2000s), we compare whether the horizontal and
vertical patterns of climatological seasonal variations can reproduce the patterns
captured by the satellite and the snapshot observations. Especially, we focused on the
Chl and PP patterns, which strongly reflect effects of the different assumptions about
how growth rates depend on light and nutrients." at the beginning of the Observational
Data section.
**1.      Lastly, an explanation of limitations and what still needs to be improved from**
**these models can be useful.**
In this study, we compared the two models, each with only one phytoplankton type: the
FlexPFT incorporating variable C:N:Chl ratios and photoacclimation, and the
InFlexPFT assuming constant composition without photoacclimation. With the
FlexPFT, a single phytoplankton type adjusts its growth rate (i.e., acclimates)
depending on available nutrients and light conditions. On the other hand, the InFlexPFT
does not account for this physiological flexibility, and therefore either light or nutrient
limitation tends to reduce growth rates more with the InFlexPFT compared to the

FlexPFT. Also, we ignored other biological processes (e.g., interactions between grazers
and phytoplanktons, export and recycling) of BGC model.
New Lines 480-482: We have revised the text to read, "In addition, we will proceed
with research on introducing flexible physiology to the growth of multiple
phytoplankton, as well as associated food quality effects on predation by zooplankton,
and the uncertainty of other biological processes, such as nitrification, grazing,
mortality, export and recycling." in the last paragraph of the Conclusions section.
----------------------------
**Technical corrections:**
**Abstract:**
**L005 - Does InFlexPFT also incorporate photoacclimation?**
New Lines 6-9: The InFlexPFT control model does not incorporate photoacclimation
because the Chl:C ratio (Eq. 5 in old version) is fixed by the equation of light limitation
(fixed stoichiometry). We revised the text to read, "We compare modeled chlorophyll
and primary production from an inflexible control model (InFlexPFT), which assumes
fixed carbon (C):nitrogen (N):chlorophyll (Chl) ratios, to a recently developed flexible
phytoplankton functional type model (FlexPFT), which incorporates photoacclimation
and variable C:N:Chl ratios." in the abstract.
**L008 - Briefly Specify where these observations are coming from.**
New Line 10: We added the "(e.g., satellite imagery and vertical profiles of in-situ
observations)" after "We coupled each phytoplankton model … and evaluate their
respective performance versus observations".
**L009 - What about nutrients? They are mentioned in the earlier line.**
We deleted "nutrients".
**L010 - Specify where this subsurface Chl maximum is reproduced, and the Chl**
**concentrations are overestimated.**
New Line 12: We add "in the subtropical gyre" after "subsurface Chl maxima".
**L014 - You should also state the role of FlexPFT incorporating photoacclimation.**
New Lines 16-18: We revised last sentence of the abstract to read: "Compared to the
InFlexPFT, the key differences that allow the FlexPFT to better reproduce the observed
patterns are its assumption of variable, rather than fixed, C:N:Chl ratios and inter-
dependent, rather than strictly multiplicative, effects of light- "(photoacclimation)" and
nutrient- "(uptake)" limitation.".
**Introduction:**

**L029-L030 - Provide further details on how they are debated.**

New Lines 32-35: "For example, some models include numerous phytoplankton and zooplankton types (Ward et al., 2013), others resolve complexity selectively for specific trophic levels (Follows et al., 2007, Gothlich and Oschlies, 2012), and others incorporate physiological trade-offs into ecological parameterizations (Smith et al., 2016, Pahlow et al., 2020)."

**L072 - cite some of the few tests that have been conducted.**

New Line 75: We added these references (e.g., Masuda et al., 2021, Matsumoto et al., 2021).

**L075 - FlexPFT is also an NPZD model no?, I would recommend rephrasing this sentence to more clearly depict the differences between the control and flexible C:N:Chl model.**

New Lines 84-90: The FlexPFT is a part of phytoplankton equation (Eq. A2) in the NPZD model.
We have revised as follows:
"In this study, we focus on the acclimative growth response of phytoplankton as incorporated in these models. To evaluate the performance and implications of this acclimative response of phytoplankton growth to varying light and nutrient conditions across the North Pacific Ocean, we compare modeled chlorophyll and primary production from an inflexible phytoplankton control model (InFlexPFT), which assumes fixed C:N:Chl ratios (fixed stoichiometry), to a recently developed phytoplankton model   (FlexPFT, Smith et al., 2016), which incorporates photoacclimation and variable C:N:Chl ratios. We apply these two phytoplankton models in a 3-D eddy-resolving ocean circulation model of the North Pacific, to assess each model's performance compared to observations of chlorophyll and primary production."

**Methods and Materials:**
**L085 - Very descriptive, but this sentence is a bit hard to follow, I would recommend restructuring to make it more clear.**

New Lines 94-104: We have revised the text to read:
"We used a coupled physical-biological model of the North Pacific, consisting of the physical ocean model, which is an eddy-resolving (1/10) OFES2 (the Ocean general circulation model For the Earth Simulator) including sea-ice (Masumoto et al., 2004, Komori et al., 2005, Sasaki et al., 2020) coupled with a simple nitrogen-based Nitrate-Phytoplankton-Zooplankton-Detritus (NPZD) pelagic model (Sasai et al., 2006, 2010, and 2016). The OFES2 domain extends from 20S in the South Pacific to 68N in the North Pacific and from 100E to 70W. The OFES2 has 1/10 horizontal resolution with 105 vertical levels, from 5 m thickness at the surface to 300 m thickness at the maximum depth of 7500 m. The physical fields were spun up for 50 years under

| 894 | climatological forcing data (wind stresses, heat flux, and freshwater flux) from the |
| 895 | Japanese 55-year Reanalysis (JRA55-do) (Tsujino et al., 2018) and from the initial |
| 896 | condition of the observed climatological fields of temperature and salinity (World |
| 897 | Ocean Atlas 2009, WOA09) (Antonov et al., 2010, Locarnini et al., 2010) without   no |
| 898 | motion for 50 years. After 50 years of spin-up integration, the OFES2 was forced by 3- |
| 899 | hourly JRA55-do from 1958 to 1979. The last day of 1979 is used for the initial |
| 900 | physical fields for performing coupled physical-biological model simulation.". |
| 901 | |
| 902 | **L101 - state the value of this initial nitrogen N field if possible, otherwise be more** |
| 903 | **specific on what you mean here.** |
| 904 | |
| 905 | New Lines 112-115: We revised from "the WOA09" to "the observed annual |
| 906 | climatological values of WOA09 The initial N concentration range from 5 to 20 (mmol |
| 907 | N m^-3) in the subpolar surface and 0.1 to 5 (mmol N m^-3) in the subtropical surface." |
| 908 | |
| 909 | **L102-L03 - is there a reason why these values were used? Add citation, reasoning,** |
| 910 | **or state that it is part of model calibration?** |
| 911 | |
| 912 | New Lines 117-118: We added some citations. We added "These P, Z, and D initial |
| 913 | values are taken from Sasai et al. (2006, 2010, 2016).". |
| 914 | |
| 915 | **L104 - This sentence feels a bit out of place. I would add this to your previous** |
| 916 | **description in L093.** |
| 917 | |
| 918 | This sentence leads to the previous section (explanation of the steady state of physical |
| 919 | fields), so we revised it as follows: |
| 920 | |
| 921 | New Lines: We revised from "The last day of 1979 is used for the initial physical fields |
| 922 | for this simulation." to "The last day of 1979 is used for the initial physical fields for |
| 923 | performing coupled physical-biological model simulation." in [L93]. |
| 924 | |
| 925 | New Lines 118-119: We revised from "Two NPZD models are incorporated after the |
| 926 | last day of 1979 of the OFES2." to "Two NPZD models are incorporated after the last |
| 927 | day of 1979 of the physical fields in the OFES2" in [L104]. |
| 928 | |
| 929 | **L115 - If Q is a function of I, N, and T, I would add that in Eq1. Q(I,N,T).** |
| 930 | |
| 931 | We added Q(I,N,T) and fv(I,N,T) are a function of I, N,T in equations in Section 2.2. |
| 932 | |
| 933 | **L116-L117 - Add citation directing to Eq.4. Fv is repeated in L125.** |
| 934 | |
| 935 | We added citation directing for each equation. |
| 936 | |
| 937 | **L124 - Explain how you determine potential maximum affinity for N. Also cite** |
| 938 | **table 1.** |
| 939 | |

These parameter values were determined by tuning the model to reproduce the seasonal and spatial variability of N and Chl in the near-surface of the North Pacific.

New Lines 178-180: These parameter values were determined by tuning the model to reproduce the seasonal and spatial variability of N and Chl in the near-surface of the North Pacific. We revised "Parameter values, mu_max, V_0, A_0, and alpha (Table 1) used in Eqs. 1 to 7 for the phytoplankton growth rate were tuned, separately for each coupled model, to reproduce the seasonal variability of N, and Chl in the near-surface of North Pacific.". after explanation of equations.

**L131- cite table 1 after the theta explanation.**

New Line 133: We have revised to cited table 1 after biological parameters used (potential maximum growth rate, potential maximum uptake rate for N, potential maximum affinity for N, and initial slope of growth versus light intensity).

**L132 – Is there a reasoning behind the activation energy Ea used? If so, cite it. Is it derived from observations?**

In this study, we set to constant Ea value for growth rate, corresponding to an doubling of rate for a 10 degree C increase in temperature (i.e., $Q_{10}$ = 2.0, which is a typical empirically-based value for the temperature sensitivity of phytoplankton growth rates (Eppley, 1972, Bissinger et al., 2008). Because it doesn't depend on phytoplankton metabolic rates under different nitrate limitation (Maranon et al., 2018, https://doi.org/10.1038/s41396-018-0105-1).

New Lines 145-148: "$E_a$ is the activation energy ($4.8 \times 10^4$ J mol$^{-1}$), which is set to a constant value, corresponding to a doubling of growth rate for a 10 degree C increase in temperature (i.e., $Q_{10}$ = 2.0), which is a typical empirically-based value for the temperature sensitivity of phytoplankton growth rates (Eppley, 1972, Bissinger et al., 2008).".

**L135 – I understand why $\mu_{InFlex}$ and $\mu_{Flex}$ are used, but they are quite lengthy, if possible I would abbreviate them to have shorter names.**

We shortened the letters. Mu_InFlex to "mu_IFL" and mu_Flex to "mu_FL" in Section 2.2 and Appendix.

**L138 - since you already explained the potential maximum uptake rate and the potential maximum affinity for N above, I don't think you need to explain them again here, but do add the last part of this sentence and citations (L139) in L124.**

We have deleted the explanation of the potential maximum uptake rate and the potential maximum affinity for N in this sentence.

**L154 - This part is difficult to follow. Expand further on this paragraph. All these parameters are introduced, but no equation explains where they come from.**

New Lines 193-198: We have revised the text to the following:
"For the InFlexPFT, Chl concentration (mg m^-3) is P (mmol N m^-3) x the constant Chl:N ratio (1.59 g Chl (mol N)^-1), and PP (mgC m^-3 day^-1) is mu_IFL P (mmol N m^-3 day^-1, Eq. 1) x the fixed C:N ratio (Redfield ratio = 106:16 mol C (mol N)^-1). In the FlexPFT, the Chl concentration (mg m^-3, = P x theta /Q) is the phytoplankton concentration, P (mmol N m^-3), x the variable Chl:N ratio (g Chl (mol N)^-1, theta /Q), and Primary Production, PP    (mgC m^-3 day^-1), is mu_FL P (mmol N m^-3 day^-1, Eq. 4) x the variable C:N ratio (mol C (mol N)^-1), 1/Q) (Eq. 5 and Smith et al., 2016)."

**L158-L163 More explanation/rationale is needed here on model evaluation and why these observational datasets were selected.**

We added the following explanation (same response to the major comment 3-2.).
New Lines 200-205: We have revised the text to clearly state, "The last 20 years (2000-2019) average of model results were compared with satellite data, in-situ observations, and the climatological data (Chl, nitrate, and temperature). Although the model and observation periods differ somewhat, using the satellite and in-situ observation data observed during the simulation period (2000s), we compare whether the horizontal and vertical patterns of climatological seasonal variations can reproduce the patterns captured by the satellite and the snapshot observations. Especially, we focused on the Chl and PP patterns, which strongly reflect effects of the different assumptions about how growth rates depend on light and nutrients." at the beginning of the Observational Data section.

**L162-170 - It would be nice to map the observations and add them as a supplementary figure. It will be easier to understand what observations you are using.**

The compared observation lines and stations are summarized in Figure 1. Two transect lines (JMA and JODC observation lines) are shown in Figures 1c, 1g, and 1k in summer map. Three time series stations (Station K2, Station S1, and Station ALOHA) are shown in Figures 1d, 1h, and 1l in winter map.

**Results and Discussion:**

**L174 - cite the satellite imagery and in-situ observations.**

New Lines 220: We revised from "the satellite imagery and in-situ observations." to "MODIS-Aqua imagery and vertical profiles of in-situ observations (JMA and JODC ship observation lines).".

**L174-177 - Should this physical evaluation go on the results. Was this part of this project or evaluated elsewhere? If so, state that.**

New Lines 225-226: We added "In addition, the seasonal variability of T and N fields in the near-surface over the North Pacific are also well reproduced (not shown)." between "The eddy-resolving ocean … mesoscale eddies, and upwelling events." and "These physical processes …".

New Lines: In the surface Chl pattern, the contrast of Chl between two gyres, and high Chl in the coastal upwelling are clearly shown using the OFES2. We added "using an eddy-resolving (1/10) OFES2" between "Overall," and "the two models" and after "… MODIS-Aqua imagery." to "In particular, the contrast between two gyres and the coastal upwelling region more clearly than lower-resolution (e.g., 1 degree, about 100 km) models (e.g., Moore et al., 2001, Vichi et al., 2007, Follows et al., 2007, Gothlich and Oschlies, 2012) by using the OFES2.".

**L182 - Throughout the manuscript, the focus is on comparing biomass and primary production between these two models, but now through the results the focus changes to comparing the chlorophyll pattern which is a proxy to biomass, but not biomass.**

We changed from "biomass" to "chlorophyll". We changed the manuscript title to "Physiological flexibility of phytoplankton impacts modeled chlorophyll and primary production across the North Pacific Ocean"

**L185 - The title should state this is a comparison since the paragraph concentrates on the model to satellite imagery comparison.**

New Lines 239: We changed the title to "Comparison of Surface Chl Patterns".

**L187- Are there any biased statistics to see how well the seasonal variations compare and what the deviations are?**

The seasonal variations and their biases are presented below the figure. The distribution is difference in chlorophyll concentration between season and climatological mean. The FlexPFT displays a similar pattern with the MODIS. Compared with the FlexPFT, the InFlexPFT shows a weak seasonal variation. In this study, we check the difference in the seasonal variations of the surface chlorophyll reproduced by the two models for comparison of MODIS, and not show the biases. In the future, we will discuss which process (physics, chemistry, and biology) contribute to the deviations and their biases reproduced in the model.

[Figure]

**L190 - More quantitative information on this model to satellite imagery comparison would be useful to understand the degree of variation.**

We added the comparison of quantitative information between MODIS and two models in this section.

**L200 - Same comment as L185 (state that it is a comparison in the title).**

New Line 261: "3.2 Comparison of Vertical Distributions of Chl and PP along the Two Transects Lines"

.

**L200 - This section is difficult to follow. I would suggest restructuring and incorporating tables or diagrams summarizing the major findings, and categorizing the different areas you are comparing.**

We revised the results section. See attached file.

**L217 - These last two sentences are a bit hard to follow, I suggest utilizing more quantitative comparisons between model and observations, to understand the degree of variation.**

We added the comparison of quantitative information between in-situ observations and two models in "Results" section.

**L269-271 - Is there reasoning why you think both models predict higher growth**
**rates here?**
New Lines 335: At 25N, in the surface, N concentration is close to zero. We added ",
high I and T enhance the growth rate" after "… despite low N concentration".
**L277-L278 - By what degree more so for FlexPFT?**
It is possible to separate and present the limitation factors. However, here we examine
the differences by simultaneously plotting the three factors (I is different symbol, N is
horizontal axis, and T is color) that control the growth rate. In addition, we explain the
effect of the C:N ratio in the FlexPFT.
T-limitation is also important around 20 degree C, which is reference temperature in T-
limitation equation, (in the subtropical region) for growth rate. Especially, in the
subsurface layer (50m in below figure), Figures 6e and 7e show the higher growth rate
compared with the that of surface layer (same N concentration and strong light
intensity). Figure (e) (165E, 35N), where is the boundary between two gyres, shows the
high growth rate around 20 degree C in the subsurface layer (50m). At other locations,
the effect of T-limitation for growth rate is smaller than the N- and light-limitations.

[Figure]

In the east-west line, the similar pattern shows below. As the locations in Figures (d)
and (e) are close to (165E, 35N), and the FlexPFT growth rate is the highest around 20
degree C.

[Figure]

**L327 - Do you mean that the spring bloom occurs across latitudes and longitudes?**
New Lines 408-410: We added "At the gyre boundary, in addition to the surface,
primary production is greater compared to other regions. Because the nutricline depth
(close to the base of the euphotic layer) and the light intensity are optimal for the spring
production." after "… the spring bloom occurs both horizontally and vertically.".
**L332- L335 - Explain why FlexPFT predicts this.**
New Lines 418-420: We added "Compared with the seasonal variations of PP in the
InFlexPFT, the FlexPFT's growth rate and the variable C:N ratio have a great influence
on the spatiotemporal variations of PP (Figs. 4, 5, and 8)." after "… coastal upwelling
region off California.".
**L345 - Chl:C instead of Chl;C**
Yes, it's a mistake. Corrected from "Chl;C" to "Chl:C".
**L346 - I think this paragraph should go earlier.**
We changed the order of this section and the previous section.
Old Figure numbers 8 and 9 changed to new figure numbers 9 and 8.
**Conclusions:**
**L376 - I think you should say you compared Chlorophyll instead of biomass.**
We revised from "phytoplankton biomass" to "Chl".
**Figures:**

**Figure 1.**
- **Minor point, but why not average from 2003-to 2019 to make the time comparison the same?**

It is 3 years shorter because we used data that deviated for a long and comparable period (MODIS imagery). It is possible to compare the same period including other sensors (SeaWiFS etc.), but only the climatological seasonal variations of surface chlorophyll are compared in this study. We confirmed, but the difference of 3 years was not so large (not shown). In the future, we plan to undertake a comparison of fluctuation components on several scales (few days to interannual).

**Figure 2.**
- **State what the white areas represent in panel a.**

We added the end of caption description. "White area in Fig. 2a is missing data.".

- **Why not use just the 2006 model year for comparison instead of 2000-2019?**

In the data from JMA, the observed location, depth, and time are different and cover a shorter period. Observations capture summer season snapshots, so chlorophyll changes significantly between north and south locations over the three months. In this study, we have compared how much the model climatological fields (smoothed fields) can explain the snapshot observation, and will investigate the impact of smaller scale process on the chlorophyll distribution, such as the snapshot, in future.

The below figure shows the daily mean chlorophyll and nitrate distributions for comparison of in-situ observation (only chlorophyll). This chlorophyll pattern in the FlexPFT model reflects the effect of small variability of vertical nitrate pattern (1 mmol N m$^{-3}$).

[Figure]

**Figure 3.**
- **I would add the text again from Figure 2. Instead of saying "same as for Fig. 2.).**

We described the same way as in Fig.2 caption.

**Figure 5.**
- **Same comment as figure 3. I would restate the information of the figure here.**

We described the same way as in Fig.4 caption.

Response to Reviewer #3

**This paper is a nice update on a line of work that aims to bring a modern representation of physiological plasticity in phytoplankton into the mainstream of ocean biogeochemical modelling. I have followed FlexPFT from a distance for a number of years, and it is useful to have a concrete illustration of how it behaves, and how it behaves differently from standard models, in a realistically complex 3D ocean simulation. I have a number of comments about how the discussion and expression of results could be improved, but I would class these as minor revisions.**

We thank Reviewer #3 for valuable comments on our manuscript. The major revised points are attached pdf file (red letters are major revised parts). Our responses (fine characters) to the Reviewer #3's comments (bold characters) are described below.

**- The review of evolving representations of phytoplankton physiology in the Introduction is especially nice.**

Thank you for your evaluation.

**- line 111: W per m^2, not W per m^3**

New Line 125: Yes, it's a mistake. Corrected from W per m^3 to W per m^2.

**- line 145-47: The role of parameter tuning in the comparison of the two model formulations is potentially very important. If the tuning of InFlexPFT had been done differently—for example, leaving mu_max the same, or lowering it further— would there have been a different pattern of similarities and differences between the 3D model runs? Which model shows a higher or lower growth rate at a particular point in space and time could be as much a matter of specific parameter choices as the structure of the equations. I would appreciate some comments on this point in the Discussion—what differences between Flex and InFlex are truly inherent and not contingent upon particular parameter choices.**

New Lines 183-186: We added the following sentences after the explanation of growth equation in the Section 2.2.
"For example, the potential maximum growth rate, mu_max, is 1.5 (day^-1) for the InFlexPFT, compared to 2.2 (day^-1) for the FlexPFT. Increasing the potential maximum growth rate decreases the surface $N$ concentration in the subpolar gyre, to the point of depleting nutrient during summer, while increasing the surface Chl concentration across the whole gyre.".

**- line 191: the success of FlexPFT at reproducing chl patterns seems to be largely a matter of dynamic range. Van Oostende et al. 2018 (https://www.sciencedirect.com/science/article/pii/S0079661117302586) also addressed this challenge in the North Pacific and found a solution by extending a standard 2-phytoplankton NPZD-style model to 3 phytoplankton compartments. So**

**perhaps the poor relative performance of InFlexPFT is really highlighting the limitations of a 1-phytoplankton model. I think this requires some discussion (in the Discussion). If one is going to improve on inflexible plankton models by adding state variables, why add them in the form of flexible physiology instead of additional fixed-response phytoplankton compartments / functional groups? There is more at stake than simple statistical performance; to me the real issue is whether we think that the ocean achieves its wide dynamic range through acclimation and plasticity, or competitive exclusion.**

In this study, we used one phytoplankton incorporated variable C:N:Chl ratios and photoacclimation. One phytoplankton in our model can handle multiple growth rates depending on available nutrients and light conditions. This corresponds to the ratio calculated by the BGC models consisting of multiple phytoplankton compartments (> 2 phytoplankton). Therefore, it is considered that the chlorophyll distribution in our model similar to the BGC model (Van Oostende et al., 2018) for calculating multiple phytoplankton with a fixed ratio was reproduced.

In general, multiple phytoplankton and zooplankton adapt to their environment and survive (e.g., growth, grazing by zooplankton, and export). Where to spend the cost is important in modeling (e.g., increasing compartments of BGC, and simplified BGC). In particular, it is important to reduce the simulation cost for the global BGC model, such as Earth System Model. In this study, we showed how much one plankton can reproduce. As future research, it is possible to introduce flexible physiology to the growth of multiple phytoplankton, or to predation by zooplankton.

**- line 245: is there any way to make this comparison with observed variation in C:N more quantitative, or at least more specific? FlexPFT seems to show about four-fold variation in C:N over a vertical profile, if I am reading the results correctly—based on the references given in these lines, does this seem like roughly the right amount of variation, or too much?**

New Lines 454-462: We added this sentence in the Discussion section (new).
"Observed elemental ratios of phytoplankton and particulate organic matter in surface layers deviate substantially from the Redfield ratio (e.g., Goldman et al., 1979, Garcia et al., 2018b, Liefer et al., 2019). Molar C:N ratios of phytoplankton vary from 4 to 60 (mol C: mol N) between phylogenetic groups. However, our current implementation FlexPFT model includes only one phytoplankton type, which varies its C:N ratio as its acclimates to the varying light, nutrient and temperature levels. Sauterey and Ward (2022) found that nitrogen and temperature mostly determined variations in the C:N ratio of phytoplankton, with variable contributions from other factors across the North Atlantic. By contrast, we find that with the FlexPFT model applied to the North Pacific, nitrogen and light levels are the primary determinants of C:N:Chl ratios, with temperature playing a lesser role. Accurate modelling of Chl and PP patterns requires accounting for these dependencies to resolve the variable elemental composition and pigment content of phytoplankton."

**- line 369: this feels like a weak comparison. What fraction of global PP _should_ the North Pacific account for? There is no additional information here, relative to**

**Fig 9, on whether FlexPFT is a quantitative improvement over InFlex. Surely there are published estimates somewhere of North Pacific PP?**

We deleted the sentence "On the other hand, … 10% of the estimated global primary production." compared with global estimation of PP.

Direct comparison between 3-D BGC model output and snapshots observed data at time series stations is difficult. In particular, the quantitative discussion is limited to the comparison of the temporal variation of the vertical profiles because it includes the influence of the uncertainty contained in the BGC model and the error due to the observed time and sea area (including depth). In this study, we are focusing on the difference in the reproducibility of the PP vertical profile between two models. The model itself still needs to be improved for quantitative discussion, so this is a topic for the future.

---

## Referee Report (RR1)

General Comments

The updated manuscript "*Physiological flexibility of phytoplankton impacts modeled chlorophyll and primary production across the North Pacific Ocean*" by Y. Sasai and colleagues highly improve the clarity and readability of the manuscript. However, I have found several grammatical mistakes and incomplete sentences that should be looked at before submission. The authors have responded to my reviewed concerns, and I am happy to recommend this manuscript for acceptance with minor revisions, as it provides valuable results that are important for the future implementation of plankton community models. However, authors should go over the manuscript and correct any grammar mistakes.

Minor comments:
L80 – remove "of"
L80 – add "a" after have
L80 – change food-web to "food web" to be consistent with how you write it in the introduction [L23]
L81 – "dissolved" instead of dissolve
L94 – replace "the" to "a" before physical ocean model?
L95 – replace including to "that includes"
L104 – add "the" after performing and before coupled
L125 – specify what "Its" is referencing.
L161 – add "the" after is and before potential uptake rate
L409 – "Because the nutricline depth …" is not a full sentence, please correct.
L415 – "," after 9c
L430 – correct suporal to "subpolar"
L450 – remove comma before and
L452 – remove space after Smith
L457 – change its to it
L457 – add comma after nutrient
L462 – "With respect to food …" this is not a complete sentence. Please correct.
L478 – add comma after future
L482 – comma before "and recycling"

---

## Author Response (AR2)

We thank two anonymous reviewers for their valuable comments on our revised manuscript entitled "Physiological flexibility of phytoplankton impacts modeled chlorophyll and primary production across the North Pacific Ocean". Our response to each reviewers' comment are as follows:

**Comments to the author**:

Response to Reviewer #1

**Referee #1**
**In my opinion, the revised version of the manuscript \*Physiological flexibility of phytoplankton impacts modeled chlorophyll and primary production across the North Pacific Ocean\* [egusphere-2022-91] solved the major concerns and accounted for the criticism raised during the first round of revisions. I appreciate the effort of the authors to answer all my questions and clarify all doubts. In this round, I read again the manuscript and collated a list of minor suggestions below. One of the suggested edits ---to split the Results and Discussion section--- resulted in a thorough Results section but a very short Discussion. I think the authors have the opportunity to further comment and clarify their results and frame them in the wider context of BGC modelling efforts nicely described in the introduction and to comment on the relative importance of photoacclimation vs alternative mechanisms in shaping the DCM. My recommendation is thus to conduct a minor revision.**

We thank Reviewer #1 for valuable comments on our revised manuscript. The discussion section has been expanded as pointed out by the reviewer #1. Our response corresponds to each comment.

**\*Minor comments\***

**L001 - change ", and hence biomass responds to changes in" to varies with"**

Line 1: We changed from "… change …" to "… to vary with light and nutrient …".

**L002 - "... capture \*variable chlorophyll:carbon ratios due to\* photoacclimation, i.e. ..."**

Line 2: We revised "… capture variable chlorophyll:carbon ratios due to photoacclimation, …".

**L007 - recurrent, but why not just call it control model, constant PFT model o similar? [fixedPFT?]**

Line 7: We revised from "an inflexible control model (InFlexPFT)" to "an inflexible phytoplankton functional type model (InFlexPFT)". We removed "control" after "inflexible" in the body text.

**L013 - "yields faster growth rates \*that result in\* high Chl ..."**

New Line 14: We revised "… yields faster growth rates that result in high Chl …".

**L018 - add concluding statement with implications for future studies and/or message to the ocean bgc community?**

New Lines 19-20: We added "These implications suggest improvement of chlorophyll and primary production patterns in the near-surface ocean in future biogeochemical models." after "… nutrient-(uptake) limitation.".

**L086 - standard NPZD instead of inflexible?**

It only explains inflexible or flexible for phytoplankton growth rate equation. Most NPZD models only assume a fixed C:N:Chl ratios, not necessarily "inflexible" models.

New Line 88: We revised from "inflexible phytoplankton control model" to "inflexible phytoplankton functional type model".

**L164 - outline the tuning procedure?**

The biological parameters in Table 1 are changed and simulated to confirm the reproduction of the climatological observed distribution of N (WOA18) and Chl (Satellite), and the biological parameter values are determined.

New Lines 167-168: We revised "… for the phytoplankton growth rate were tuned, separately for each coupled model, to confirm the reproducibility of the climatological seasonal variability of observed N, and Chl patterns in the near-surface of North Pacific.".

**L171 - reproductivity? [I did not understand the sentence]**

We removed this sentence after "… Chl concentration across the whole gyre.".

**L185 - perhaps motivate a bit more why you focus on climatological patterns rather than point estimates (i.e. interest on large scale variation rather than short term noise, etc). In principle nothing precludes a direct comparison aligning model output with satellite data.**

New Line 188: We added "to investigate the large scale variation over the North Pacific." after "The last 20 years … and the climatological data (Chl, nitrate, and temperature)".

**L190 - Ocean color data can be cited using a DOI; e.g. see [this website](https://oceancolor.gsfc.nasa.gov/citations/)**

New Line 194: We changed data citation site "http://doi.org/10.5067//AQUA/MODIS/L3M/CHL/2022" and added it in data availability and acknowledgment sections.

**L225 - Active voice? [Figure 1 shows ...]**

New Lines 228-229: We revised "Fig.1 shows the surface Chl distribution … .".

**L238-L241 - Please simplify; in my opinion it would be ok to just highlight that FlexPFT results $\theta / Q$ ratios in 0.1-3.0 *vs* the fixed value of 1.59 in InFlexPFT.**

New Lines 243-244: We revised "… the difference between the FlexPFT's variable Chl:N ratios in 0.1 to 3.0, versus the fixed value of 1.59 in the InFlexPFT..".

**L247-L251 belong to discussion**

We moved these sentences in discussion section (New Lines: 436-440).

**L269 - better, ok, but how much better?**

New Line 266: We added ", especially, the depth and structure of SCM." after "… Chl distribution much better than the InFlexPFT".

**L280ff - please detail the nature of the figures provided, if they refer to a specific season, etc. It would be helpful to quote some integrated figures too, like those provided in the response letter. Quantities like the fraction of PP accounted for by the DCM and the thermocline are also of interest.**

New Lines 282-284: We added "In summer (Figs 2 and 3), the SCM is clearly formed in the subsurface layer except for the subpolar gyre (north of 40N). The vertical distributions of PP and phytoplankton growth rate form maxima along the nutricline depth." after "… whereas it is constant for the InFlexPFT.".

**L415 - subpolar [typo]**

Line 415: Yes, it's a mistake. We corrected "subpolar".

**L419 - it would be nice to quote here other estimates, or to mention them in the discussion ...**

New Lines 421-423: We added "Although not directly comparable to our estimates, the global primary production as estimated by the satellite and global biogeochemical models remains large: from 38.8 - 42.1 PgC yr^-1 over the period of 1998 - 2018 (Kulk et al., 2020) and from 38 - 79 PgC yr^-1 (Carr et al., 2006)." after "… in the North Pacific basin … twice that of the InFlexPFT.".

**L431 - please expand a bit ... e.g. [Moeller 2019](https://doi.org/10.1038/s41467-019-09591-2) light dependent grazing, [Wirtz 2022](https://doi.org/10.1038/s41558-022-01430-5) motility ... and a long etc. As commented above, there is room to improve the discussion, even by moving some materials from results.**

New Lines 436-446: We revised and added "It is impossible to capture the vertical profiles of Chl with satellite observations, and it is therefore important to verify the SCM field reproduced by the model using in-situ observations (e.g., Shulenberger and Reid, 1981,Furuya1990). Using a 3-D biogeochemical ocean model coupled with the same FlexPFT model, (Masuda et al., 2021) showed that the observed global scale SCM distribution can be reproduced by incorporating photoacclimation in response to varying nutrient and light conditions. Various mechanisms likely contribute differently in different oceanic regimes and other mechanisms are important for reproducing specific features of SCM (e.g., Moeller et al., 2019, Wirtz and Smith, 2020). Moeller et al. (2019) proposed a new mechanism, which is light-dependent grazing by microzooplankton reduces phytoplankton biomass near the surface but allows accumulation at depth, for SCM formation. Furthermore, vertical migration by phytoplankton can explain the occurrence of SCM consistently above the nutricline depth, which photoacclimation alone cannot (Wirtz and Smith, 2020). Wirtz et al. (2022) also suggested that phytoplankton vertical migration fuels up to 40 % (>28 tg yr^-1 N) of new production and directly contributes 25 % of total oceanic net primary production (herein estimated at 56 PgC yr^-1) using their model." after "… nutrient and light conditions.".

**L452 - please remember the comment about featuring a concluding statement in the abstract**

We added "These implications suggest improvement of chlorophyll and primary production patterns in the near-surface ocean in future biogeochemical models." in abstract (New Lines 19-20).

**L470 - please check whether it is possible to cite those dataset using a DOI**

We could not confirm a DOI. Web site is only.

**L481 - if there is a single $k$ then it is just one coefficient, and the question is whether it is sensible to assume similar diffusion in the vertical and the horizontal; otherwise, if there is a $k_{x,y}$ that differs from $k_{z}$ please update the notation ...**

New Line 489: We revised "K_h is the lateral diffusion coefficient and K_v is the vertical diffusion coefficient in the OFES2, …" .

**L489 - please comment on the need to include a linear respiration term and, especially, a quadratic mortality term ... as highlighted in the first round, the formulation is a bit conterintuitive or nonstandard ...**

The NPZD model used in this study is constructed based Sasai et al. (2016) "Coupled 1-D physical-biological model", and it differs from the widely used NPZD model. A linear respiration term may not have been necessary for our study, but it is left as is because it has a smaller effect than the other terms. By using quadratic mortality equations of phytoplankton and zooplankton, the 3-D NPZD model simulation can be stabilized (do not generate unforced oscillation) in low-concentration regions (e.g., subtropical gyre and below the euphotic layer). In the future, we will verify the uncertainty of each biological processes.

Response to Reviewer #2

**Referee #2**
**The updated manuscript "Physiological flexibility of phytoplankton impacts modeled chlorophyll and primary production across the North Pacific Ocean" by Y. Sasai and colleagues highly improve the clarity and readability of the manuscript. However, I have found several grammatical mistakes and incomplete sentences that should be looked at before submission. The authors have responded to my reviewed concerns, and I am happy to recommend this manuscript for acceptance with minor revisions, as it provides valuable results that are important for the future implementation of plankton community models. However, authors should go over the manuscript and correct any grammar mistakes.**

We thank Reviewer #2 for valuable comments and English grammar checks on our revised manuscript. Our response corresponds to each comment.

**Minor comments:**
**L80 – remove "of"**

New Line 81: We removed "of" between "Most" and "biogeochemical models".

**L80 – add "a" after have**

New Line 81: We added "a" after "have".

**L80 – change food-web to "food web" to be consistent with how you write it in the introduction**
**[L23]**

New Line 81: We changed "food web" to be consistent with previous sentence [New Line 24].

**L81 – "dissolved" instead of dissolve**

New Line 82: We revised "dissolved".

**L94 – replace "the" to "a" before physical ocean model?**

New Line 95: We replaced "a" before "physical ocean model".

**L95 – replace including to "that includes"**

New Line 96: We replaced from "including" to "that includes".

**L104 – add "the" after performing and before coupled**

New Line 105: We added "the" after "performing" and before "coupled".

**L125 – specify what "Its" is referencing.**

New Line 126: We revised from "Its" to "The phytoplankton growth rate equation".

**L161 – add "the" after is and before potential uptake rate**

New Line 154: We added "the" after "is" and before "potential nutrient uptake rate".

**L409 – "Because the nutricline depth …" is not a full sentence, please correct.**

New Lines 394-395: We revised "In this region, the nutricline depth … .".

**L415 – "," after 9c**

New Line 400: We added "," after "(Figs. 9c".

**L430 – correct suporal to "subpolar"**

New Line 415: We corrected "subpolar".

**L450 – remove comma before and**

New Line 440: We removed "," before "and other mechanisms …".

**L452 – remove space after Smith**

New Lines 444-445: We removed space after "(Wirtz and Smith)".

**L457 – change its to it**

New Line 451: We changed from "its" to "it".

**L457 – add comma after nutrient**

New Line 451: We added "," after "nutrient".

**L462 – "With respect to food …" this is not a complete sentence. Please correct.**

New Lines 456-457: We corrected "This may be important with respect to food quality effects (e.g., Kwiatkowski et al., 2018; Mastsumoto et al., 2020)".

**L478 – add comma after future**

New Line 471: We added "," after "future".

**L482 – comma before "and recycling"**

New Line 475: We added "," before "and recycling".